

# Implementation of a physically based water percolation routine in the Crocus (V7) snowpack model

Christopher J. L. D'Amboise[1,2], Karsten Müller[1], Laurent Oxarango[3], Samuel Morin[4], Thomas V. Schuler[2]

[1]Norwegian Water Resources and Energy Directorate, Oslo, 0368, Norway
[2]Department of Geoscience, University of Oslo, Oslo, 0316, Norway
[3]Univ. Grenoble Alpes, CNRS, IRD, IGE, F-38000 Grenoble, France
[4]Météo-France – CNRS, CNRM UMR 3589, Centre d'Etudes de la Neige, Grenoble, France

*Correspondence to*: Christopher J. L. D'Amboise (chda@NVE.no)

**Abstract.** We present a new water percolation routine added to the 1D snowpack model Crocus as an alternative to the empirical bucket routine. This routine is based on Richards equation, and describes flow in an unsaturated porous medium governed by capillary suction and hydraulic conductivity of snow layers. We tested the Richards routine on two data sets, one recorded from an automatic weather station over the winter of 2013-2014 at Filefjell, Norway, and a simple synthetic data set. Model results using the Richards routine generally lead to thinner and denser simulated crust layers compared to the bucket routine. Wet snow layers often reach the transition between the pendular and funicular regimes, with 17% of snow layers obtaining saturation of >10% and 4.3% of layers had saturation of >15% for the synthetic data set. The Richards routine had a maximum liquid water content of 167.3 kg m$^{-3}$ where the bucket routine had a maximum of 42.1 kg m$^{-3}$. To express the water retention curve and the hydraulic conductivity of snow layers, the Richards routine heavily relies on accurate density and grain size estimations. We found the Richards routine was sensitive to the chosen modelling time step. The time step dependency is a result of feedback between the water percolation routine and the snows compaction and metamorphism routines. We show that the new routine has been implemented in the Crocus model, but due to amplification of parameter uncertainties through a number of feedbacks, meaningful applicability is limited until new and better parameterizations of water retention is developed for different snow types.

## 1 Introduction

Knowledge about the process of water percolation in the snowpack is necessary for improving many applications such as flood forecasting, river and reservoir management, slope stability and avalanche forecasting. Measuring liquid water content (LWC) of snow layers is not practical because it is time consuming and, LWC has the ability to dramatically change over the timescales, that are considerable shorter than those of observations (Techel and Pielmeier, 2011; Wakahama, 1975). When water is introduced to a snowpack, snow stability is able to change rapidly, because cohesion strength depends on the amount of water saturation (Ambach and F. Howorka, 1966; Brun and Rey, 1987; Hartman and Borgeson, 2008). Because LWC of a snowpack can change quickly, avalanche forecasters, mountain guides, researchers and rescue workers, have reported that they do not fully trust classical snow stability tests (e.g. Rutschblock and extended column tests) when performed in wet snow





(Techel and Pielmeier, 2009). To improve flood and avalanche forecasting capabilities, detailed snowpack hydraulic information on fine spatially and temporally scales is required. One way to achieve a detailed view of snow hydraulics is to supplement meteorological and hydrological observations with physically based percolation modeling.

Vertical water flow through a layered snowpack can occur in two different modes, matrix flow and preferential flow. Matrix
flow is a diffusive flow through the pore space of the snowpack, which sets up a uniform water front. Preferential flow, also called finger flow, is when water quickly flows in channels to deeper layers (Marsh and Woo, 1984). A combination of both flow schemes occurs in a snow layer; often preferential flow will initiate wetting a dry snow layer, followed by an expansion of flow paths that will end up in matrix flow (Williams et al., 2010). As water percolates through an isothermal snowpack, preferential flow paths are created, and water transport through the snowpack becomes very efficient (Colbeck, 1979). Using
multi-color dye tracer experiments, Schneebeli (1995) has shown that the preferential flow channels in an isothermal snowpack can migrate over time.

Gravity and capillary forces govern water movement in unsaturated snow (Colbeck, 1972; Jordan et al., 1999). Capillary forces arise from adhesion and surface tension of liquid water inside the pore space of the snowpack. Snow layering produces vertical gradients since capillary pressure has an inverse relationship to pore size, (Wankiewicz, 1978). Pressure gradients acting
against the gravity may induce the formation of preferential flow channels, as flow channels in soil occur where capillary pressure gradient opposes the waters flow direction (Philip, 1975). Two common textural barriers are crust layers and neighboring snow layers with sharp grain size differences. Assessing the hydraulic conductivity of textural barriers is not straightforward, because layers behavior may vary greatly; such as crusts which can act as an impermeable layer or act similar to vertical conduits (Jordan, 1995). Fine-grained snow layered above coarser grains give rise to flow barriers due to capillary
pressure gradients opposing gravity. An area of high saturation can be found above such barriers and lateral water flow due to suction, terrain slope and water pooling are a common result (Colbeck, 1974b; Williams et al., 2010).

Physically-based models of water percolation through snow were first developed to describe gravitationally driven flow through isotropic isothermal snow neglecting capillary effects(Colbeck, 1972). Model complexity evolved and snow layering was introduced (Colbeck, 1974b, 1975), and heterogeneous flow where water is routed to deeper layers via flow channels
(Colbeck, 1979). Model development on percolation in a cold snowpack was addressed by including thermodynamics to percolation models (Bengtsson, 1982; Colbeck, 1976; Illangasekare et al., 1990). Capillary forces were introduced in some early models despite the deficiency in parameter sets (Colbeck, 1974a; Jordan, 1983; Wankiewicz, 1978). Wankiewicz (1978) concluded that more information on snow microstructure is needed to improve modeling of water percolation through a layered snowpack. Some recent models include gravity and suction driven preferential flow in isothermal snow layers (Hirashima et
al., 2014; Katsushima et al., 2009).

Measuring the hydraulic conductivity and water retention of snow layers is a time consuming task performed in a cold lab and so far cannot be conducted in the field or even deployed as an autonomous recording system. Yet, new parameterizations of the retention curve (Yamaguchi et al., 2010, 2012) and hydraulic conductivity (Calonne et al., 2012) have been developed recently. These developments have been taken advantage of in percolation models based on Darcy's law (Hirashima et al.,



2010) and the Richards equation (Wever et al., 2014, 2015). Hydraulic conductivity and water retention of snow layers give insight on how the snowpack may evolve in regards to LWC. For instance, the 2012 surface runoff anomaly from the Greenland ice sheet can be explained by the reduced hydraulic conductivity of near surface firn and ice layers. Growth of near surface ice lenses prior to the 2012-melt-season caused a low hydraulic conductivity between surface layers and deep firn and ice layers.

This effectively sealed off the available pore space in deep, cold firn which usually absorbed a large part of meltwater, thereby causing an increased amount of early season runoff (Machguth et al., 2016). Although having important consequences, few models are capable of adequately simulating the formation of ice lenses and their hydrological impact. A dual domain Richards based model has begun explaining preferential flow paths, which can reproduce some of the ice crusts present in the snowpack (Wever et al., 2016). The Richards equation when applied to snow describes water percolation through a porous ice matrix

considering the water retention curve and the hydraulic conductivity of snow layers. A one-dimensional Richards equation solver was recently added to the detailed snow model SNOWPACK (Wever et al., 2014, 2015). We added a similar physical water transport routine to the snowpack model Crocus. This paper will discuss the parameterization, sensitivity of the new routine and compare difference in water percolation with the bucket routine for two data sets.

## 2 SURFEX, ISBA and Crocus description

SURFEX is an atmosphere to surface coupling model, with ISBA (Interactions between Soil, Biosphere, and Atmosphere) being the land surface scheme (Noilhan and Mahfouf, 1996). The Crocus model (Brun et al., 1989) is the most detailed of three snowpack models embedded in the ISBA routine. It was classified in the group of "most complex snow models" by the Snow Models intercomparison project (Etchevers et al., 2004). Crocus is a one-dimensional multi-layer model that describes the snow microstructure evolution based on environmental conditions. This model simulates the snowpack from the first snow

to melt out, by calculating mass and energy fluxes between snow layers and its interface with the underlying ground and overlying atmosphere. Processes that act on and in the snowpack are summarized in Vionnet et al. (2012), which are represented in Crocus by routines (shown in Fig. 1). These routines run in a sequential manner. This paper will discuss a new option for the water percolation routine for the Crocus model but does not consider aspects of coupling between Crocus and other components in ISBA and SURFEX. For a detailed description of the implementation of Crocus in SURFEX and a

detailed description of Crocus see Vionnet et al. (2012).

The SURFEX/ISBA-Crocus default time step is 15 minutes denoted (T$_{Crocus}$), but in this study, T$_{Crocus}$ will be varied to examine sensitivity and degree of coupling to the soil. Crocus requires the following forcing variables; air temperature, humidity, wind speed, incoming shortwave and longwave radiation, solid and liquid precipitation rate and atmospheric pressure (Vionnet et al., 2012). Focing data is generally provided by highly instrumented test-sites, output from numerical weather prediction

models (Vernay et al., 2015) or an assimilation/combination of the two (Durand et al., 2009).

The water transport and refreezing processes are expressed in the SNOWCROREFREZ routine (Fig. 1). Feedbacks exist between the liquid water transport (SNOWCROREFREZ) and other processes such as snow compaction



(SNOWCROCOMPACTN) and metamorphism (SNOWCROMETAMO). It is important to realize that changes to the amount and timing of water percolation will feedback to other routines and affect other snowpack variables.

## 2.1 The bucket approach

To describe water percolation in snow, Crocus has been using an empirical based routine, the so-called bucket routine that has
been calibrated using long time series of lysimeter data (Morin et al., 2012) and drainage experiments on the irreducible water content (Coleou and Lesaffre, 1998). The bucket routine uses a holding capacity, defined by a percentage of the snow layers pore space. A snow layers "bucket" is filled up with water when water is introduced via rain or melt. Once the bucket is full, overflow occurs to fill up the subsequent snow layer's bucket, restricting water motion to downward direction. Crocus bucket routine uses 5% of a snow layers pores spaces volume as the holding capacity or "bucket size" (Vionnet et al., 2012). For
Crocus, the holding capacity is proportional to the density of the snow layer and is independent of snow grain type or surrounding environment. The size of the "buckets" is not agreed upon in literature, which is discussed in detail in Lafaysse et al. Sec. 3.7 (2017). Singh et al. (1997), found water holding capacity to be 6.8% of a snow layers volume (note this is total volume not just pore space). However, when an impermeable layer was set beneath it, the holding capacity rose to 14.2%, showing that the surrounding environment has an effect on LWC of a snow layer, at least for a short time period.

**3 Implementation of Richards routine in Crocus**

Richards routine describes motion of water through an unsaturated porous matrix considering capillary driven and gravity flow. Recent developments in parameterizing snows conductivity and snowpack water retention allow for the implementation of Richards equation in layered snow pack models. In contrast to the bucket model, the Richards routine allows for upward motion of water if capillary pressure conditions are suitable.
A Richards solver has recently been implemented in the SNOWPACK model (Wever et al., 2014). A comparison between the bucket percolation and Richards percolation in the SNOWPACK model showed that Richards routine performed better than the bucket routine for a sub-day time scale when compared to lysimeter data (Wever et al., 2014, 2015). However, Avanzi et al. (2015) found speed of water transport with the Richards equation over a capillary barrier to be underestimated when compared to experimental results, but the model reproduces an increased LWC above barriers.
This paper discusses the implementation of a similar routine in the Crocus model. The new routine SNOWCROPERCO_RCH represents an alternative to the SNOWCROPERCO routine (bucket) (Fig. 1). We use the snow layer discretization in Corcus as a mesh for solving the Richards equation.

Richards equation (eq. 1) is a non-linear partial differential equation describing the water mass balance of snow with water fluxes expressed using a generalized Darcy law taking into account the dependence of hydraulic conductivity with water
content. Its main variables are the pressure head (h) and the volumetric liquid water content (θ).

$$\frac{\partial \theta}{\partial t} = \frac{\partial}{\partial z}\left(K(\theta) \cdot \frac{\partial H}{\partial z}\right) \tag{1}$$



K(θ) is the hydraulic conductivity which is a function of the volumetric water content (θ), and t and z denote time and depth (positive downward). H is the hydraulic head, which is the sum of the pressure head (h) and the elevation (z), which is negative because z is positive downward (Eq. 2).

$$H = h - z \tag{2}$$

The water retention curve h(θ) and the hydraulic conductivity function needs to be expressed for each snow layer.

### 3.1 Water retention curve

Using the Van Genuchten (1980) parameterization, the water retention curve can be expressed with Eq. (3) if four parameters (α, n, $\theta_r$, $\theta_s$) are known. Where α and n are the Van Genuchten fit parameters (see Eq. 4 ) and $\theta_s$ (Eq. 5) and $\theta_r$ (Eq. 7) are the saturated water content and the residual water content, respectively.

$$\theta = \theta_r + (\theta_s - \theta_r) \cdot (1 + (\alpha \cdot h)^n)^{-\left(1 - \frac{1}{n}\right)} \tag{3}$$

Recent experiments (Yamaguchi et al., 2010, 2012) and theoretical estimates based on prior experiments (Daanen and Nieber, 2009) propose parameter sets for these four variables. This paper utilizes the Yamaguchi et al. (2012) parameter set (we also provide options in Crocus to use two alternative parameter sets, see Appendix B).

$$\alpha = 4.4 \times 10^6 \cdot \left(\frac{\rho_{snow}}{D \cdot 1000}\right)^{-.98} \tag{4}$$

$$n = 1 + 2.7 \times 10^3 \cdot \left(\frac{\rho_{snow}}{D \cdot 1000}\right)^{0.61} \tag{5}$$

Where $\rho_{snow}$ is the dry density of the snow, D is grain size diameter and P is porosity (volume of pore space).

$$\theta_s = 0.9 \times P \tag{5}$$

Yamaguchi et al. (2012) performed a drainage experiment to obtain the Van Genuchten fit parameters. The gravitational drainage experiment assumed that the saturated hydraulic conductivity $\theta_s$ for snow to be 90% of the pore spaces. This is due

to small air bubbles that become trapped in the pores between the snow grains as the snow saturates. One should note that Yamaguchi's study examined samples of melt form and small rounded grains with a density range of 361- 636 kg m$^{-3}$ and grain size range of 0.05 mm to 5.8 mm. Columns of snow were saturated with 0˚C water and left to drain. The study found a parameter set for melt form crystals and concluded that rounded crystals could not be represented with the same parameter set as melt forms.

The Van Genuchten parameterization being applied is adopted from soil science for flow in an unsaturated soil matrix. The residual water content in snow presents specific challenges that are not present when applied to soil. Snow is often completely dry via phase transform, where soil is assumed to always have a small amount of liquid water. $\theta_r$ is defined as the amount of water that remains in the porous medium with infinite suction being applied. It corresponds to disconnected water patches entrapped in the pore system. Following Yamaguchi et al (2010), a residual water content $\theta_r$ =0.02 is adopted. However, the

LWC of a snow sample can further 'dry out' via evaporation and freezing, resulting in a negative saturation (S) (Eq. 6).

$$S = \frac{\theta - \theta_r}{\theta_s - \theta_r} \quad \text{Where } \theta_r < \theta < \theta_s \tag{6}$$





Negative saturation (where $0 \leq \theta < \theta_r$) is physically possible with phase change, but causes numerical problems. Therefore, saturation needs to be restricted between 0 and 1. To overcome this limitation we use a continuous piecewise function to keep $\theta > \theta_r$ (Eq. 7).

$$\theta_r = \begin{cases} .02 & if \quad \theta > .02 \\ .75 \cdot \theta & if \quad \theta < .02 \end{cases} \tag{7}$$

To avoid infinite values of the hydraulic head (Fig. 2), and hydraulic conductivity (Fig. 3, Sec. 3.2) of 0 when approaching LWC=0 a small amount of water needs to be added to snow layers that are completely dry. In this study we call this added water 'pre-wetting' or $\theta_{min}$. Pre-wetting of volumetric water content $\theta_{min}$ = $10^{-6}$ (unitless) (as default, but a sensitivity study varies this amount) is added to layers that have $\theta < \theta_{min}$ to keep the hydraulic conductivity value numerically different from zero. To conserve mass and enthalpy of the snow pack the water added to a dry ($\theta < \theta_{min}$) layer is melted from the layers

ice/snow, keeping the density of the layer unchanged. When this is done the snow layers temperature is cooled according to the amount of latent heat required to melt ice corresponding to $\theta_{min}$. This means that in this step, our implementation allows the existence of liquid water, even at temperatures below 0ºC. Nevertheless, this is a technical maneuver to avoid runaway head values and conductivity values approaching zero for dry snow layers, and after the water motion is described, the water balance is closed again by refreezing $\theta_{min}$ if required by temperature conditions.

**3.2 Hydraulic conductivity**

Hydraulic conductivity (K) is a function of water content ($\theta$) see Eq. (8), as a snow layer gets wetter the conductivity will increase (Fig. 3).

$$K(\theta) = k_r(\theta) \times K_{sat}(D, \rho_{snow}) \tag{8}$$

The Van Genuchten-Mualem equation (Van Genuchten, 1980 and Eq. 9) is used to calculate the relative permeability kr. It is

implemented as a function of h (using Eq. 3 for the relation between h and $\theta$):

$$k_r = \left(1 + |\alpha \cdot h|^{\frac{1}{1-m}}\right)^{\frac{-m}{2}} \cdot \left(1 - \left(1 - \left(1 + |\alpha \cdot h|^{\frac{1}{1-m}}\right)^{-1}\right)^m\right)^2 \tag{9}$$

Hydraulic conductivity reaches a maximum when the snow matrix is saturated, known as conductivity at saturation ($K_{sat}$). Since conductivity at saturation is dictated by snow structure, which is a complex system it is described by a simple statistical model in both parameters sets available. It should be noted that both $K_{sat}$ (Eq. 10) and $k_r$ (Eq. 9 through $\alpha$ and m) are dependent

on the dry density of the snow and the grain size.

Calonne et al (2012) used 3D images of snows microstructure to derive the conductivity from different snow types and densities ranging from <100 kg m$^{-3}$ to ~550 kg m$^{-3}$, seen in Eq. (8). Equation 8 is the preferred parameter set due to the range of snow types and densities that went into deriving the equation (other alternatives described in appendix B).

$$K_{sat} = 3.0 \cdot \left(\frac{D}{2}\right)^2 \cdot exp(-0.013 \cdot \rho_{snow}) \cdot \left(\frac{G \cdot \rho_{water}}{\mu_{water}}\right) \tag{10}$$

G is gravity 9.816 m s$^{-2}$ and $\mu_{water}$ =0.001792 kg m$^{-1}$ s$^{-1}$ the dynamic viscosity of water at 0°C.



### 3.3 Solving Richards equation

To solve Richards equation, we utilize the following strategy: A finite volume discretization is applied taking each snow layer as integration volume. The average pressure head of each snow layer (corresponding to its LWC) is supposed to apply in the center of the layer.

The water fluxes ($\Phi$) are computed at the interface between layers. Counted positive when entering a snow layer, the flux at the top and the bottom are computed respectively as:

$$\Phi_{i,top}^{t+1} = K_{top}^{t+1}(\theta_i^{t+1}, \theta_{i-1}^{t+1})\left(\frac{h_{i-1}^{t+1}-h_i^{t+1}}{\Delta z_{top}} + 1\right) \tag{11}$$

$$\Phi_{i,bot}^{t+1} = K_{bot}^{t+1}(\theta_{i,}^{t+1}, \theta_{i+1}^{t+1})\left(\frac{h_{i+1}^{t+1} - h_i^{t+1}}{\Delta z_{bot}} - 1\right)$$

Where i is the index for snow layer (layer 1 at top of the snow pack). Values on the interface between two snow layers are

indicated with *top* and *bot* (bottom) subscript respectively.

$K_{top}$ and $K_{bot}$ are the hydraulic conductivity of the upper and lower boundary interfaces of layer *i*, respectively. To compute $K_{top}$ and $K_{bot}$ the arithmetic mean was used as an estimate of the conductivity at snow layers interfaces; shown in Eq. (10).

$$K_{top} = \frac{K_i \cdot \Delta z_i + K_{i-1} \cdot \Delta z_{i-1}}{\Delta z_i + \Delta z_{i-1}} \qquad K_{bot} = \frac{K_i \cdot \Delta z_i + K_{i+1} \cdot \Delta z_{i+1}}{\Delta z_i + \Delta z_{i+1}} \tag{12}$$

$\Delta z_{bot}$ and $\Delta z_{top}$ is the distance between layer mid points as described below in Eq. (11).

$$\Delta z_{top} = \frac{\Delta z_i + \Delta z_{i-1}}{2} \qquad \Delta z_{top} = \frac{\Delta z_i + \Delta z_{i+1}}{2} \tag{13}$$

Averaging conductivity of two adjacent snow layers with a simple arithmetic mean as an estimate for the interface value may be over simplifying the snows conductivity. Snow grain size, density and crystal types often have sharply defined boarders. These parameters would have great influence on the snow hydraulic conductivity. It could be argued that a piecewise function comprised of individual snow layer's conductivity better describes the vertical pattern of conductivity in the snow pack

(Szymkiewicz, 2009). However, when a piecewise function was tested, "dry layers" (with $\theta_{min}$) caused impermeable barriers and caused numerical problems for the simulation. Other options for combining conductivity values of snow layers to estimate the interface are possible, but more research is needed to understand how the interface conductivity behaves.

The time discretization is a Crank-Nicolson finite differences scheme, which is second order accurate in time. The non-linearity of the equation is then dealt with the iterative methodology proposed by (Celia et al., 1990). It approximates $\theta_i^{t+1,k+1}$ by a

truncated Taylor series Eq. (14).

$$\theta_i^{t+1,k+1} = \theta_i^{t+1,k} + \frac{d\theta}{dh}\Big|_i^{t+1,k}\left(h_i^{t+1,k+1} - h_i^{t+1,k}\right) \tag{14}$$

Where the superscript *k* refers to the evolution of the iterative process. $\frac{d\theta}{dh}\Big|_i^{t+1,k}$ is the derivative of the retention curve (Eq. 3) computed analytically for $\theta_i^{t+1,k}$ (or $h_i^{t+1,k}$). The final discretized form of Richards equation (Eq. 1) including the Crank-Nicholson scheme with Celia decomposition reads as:





$$\frac{\theta_i^{t+1,k}+C_h^{t+1,k}\left(h_i^{t+1,k+1}-h_i^{t+1,k}\right)-\theta_i^t}{\Delta t}\Delta z_i \;= 0.5\left(\Phi_{i,top}^{t+1,k+1}+\Phi_{i,bot}^{t+1,k+1}\right) + 0.5\left(\Phi_{i,top}^t+\Phi_{i,bot}^t\right) \qquad (15)$$

The system of discretized equations is solved with respect to the pressure head *h* at each computation step. The three-diagonal linear system is solved with the direct LU Thomas algorithm. The value of the volumetric content is then updated until the convergence criterion of the Picard iterative process is reached (see appendix A).

The Richards equation is solved on a variable time step denoted with a $\Delta t$ (see appendix A for how time step varies). The SURFEX/Crocus model run on a fixed time step $T_{Crocus,}$ which is set to 15 minutes unless otherwise specified. Snow layer properties such as grain size and snow density and snow layer temperature are updated on each $T_{Crocus}$. The variable time step segments $T_{Crocus}$ up into smaller steps as needed, until $\Sigma \Delta t = T_{Crocus}$. Outflow to soil and liquid water content of each layer are updated in the main Crocus routine when the time steps join. Finally, density, snow temperature and liquid water content are updated and the Crocus routine is finished.

## 3.4 Boundary conditions

The rain rate and evaporation rate are imposed as a flux for the upper boundary condition. For the snow-atmosphere interface rain/evaporation rates replace $\left(\Phi_{i,top}^{t+1,k+1},\Phi_{i,top}^t\right)$. Both rain and evaporation fluxes are provided from the meteorological forcing. The lower boundary is the soil snow interface. Properties from the upper most soil layer are imported from the Surfex/ISBA soil routine, which also uses the Richards equation for unsaturated water flow. Interfacial conductivity between the soil and bottom snow layer $K_{bot}$ is calculated with Eq.12 using the top soil layer hydraulic conductivity and thickness. The top soil layers pressure head and thickness are used in Eq, 10 to calculate flux to soil $\Phi_{i,bot}^{t+1}$. Both the soil properties and flux at snow pack surface remain constant over the routine inner time steps and are updated on the time step ($T_{Crocus}$). The bucket model is used for melt out, when there is less than 0.05 m of snow.

## 4 Forcing data / experiments

The Richards routine was tested on two data sets: one from Filefjell, Norway (61.178231 N, 8.112925 E) recorded at an automatic weather station by the Norwegian Water and Energy Directorate (NVE) and the other is a synthetic data set.

## 4.1 Filefjell

The Filefjell data set was recorded at hourly steps over the 2013-2014 winter at a flat field at 956 m a.s.l. Filefjell is located about 200 km North West of Oslo. Despite being 30 km inland from the end of a fjord, Filefjell is considered to have a continental snow climate, which has an averages precipitation of 603 mm year[-1] and large annual temperature variation shown in Fig. (4). Continental snow climates are characterized by thin snow covers, cold temperatures and few rain on snow events during the winter months (McClung and Schaerer, 2006). Temperatures become as cold as -28.3 °C and after January, there is



a long cold spell where temperatures stay well below zero for about a month. The surface incident shortwave radiation is low during winter, where maximum daily values are below 300 W m$^{-2}$ from October 18, 2013 to February 19, 2014, due to the high latitude. The first large rain on snow event occurs on March 7, 2013, with a second event on April 6 & 7, 2013. Early winter rain on snow events occurred before January, which is not typical for continental climates. Unfortunately, no manual snow pit measurements at the field site have been conducted during the winter of 2013-2014. However a snow pit observation was recorded March 19, 2014 approximately 3 km away on a 24° south east facing slope (Fig. 5) (Solemsli, 2014).

## 4.2 Synthetic

Figure 6 shows the 90-day synthetic data set. The peak radiation values for this data set are low and increase linearly from 100 to 200 W m$^{-2}$. Temperature has a linear increase from 265°K to 276°K. The temperature and radiation patterns set in the synthetic data set were chosen to induce a modest melt rate early in the simulation that is ramped up to a heavy melt rate at the end of the simulation. The synthetic data set is designed to test the new routines for a large range of water supply rates. This data set has two large snow events the first one starts on day 3 and deposits 2.6 m over five days. The second event occurs on day 60 after the first snow event had a chance to settle down to 1.1 m and the diurnal radiation cycle has a chance to form a melt freeze crust at the surface. The second snowfall, did not only bury the old surface crust under 1.8 m (for Richards routing, 1.9 for bucket routine) of snow, which is deposited with density variations that result from the radiation cycle. The synthetic data set allows for a simple comparison between the Richards routine and bucket routine without complicated snowpack structures, over a wide variety of melt intensities.

## 4.3 SURFEX configuration of the model runs

The C13 routine was used for snow metamorphism in the presented simulations, which uses optical diameter and sphericity to describe snow microstructure (Carmagnola et al., 2014). The soil layer sizes have been modified from the default values to allow for convergence during periods of intense water input. For the Filefjell simulation the top soil layers was 1m thick and the simulated data set had a 0.3m soil layer. Thickness of the top soil layer was adjusted as needed to allow for convergence during periods of high water flux between snow and soil.

## 5 Results

### 5.1 Filefjell Simulation

Simulation results for the Filefjell data set are presented in Fig. (7) with the top (7 A & B) showing amount of liquid water and snow layer density from the simulation run with the bucket routine and bottom (7 C & D) with the Richards routine. These simulations used T$_{Crocus}$ =15 min, but the resolution of Figure 7 is 3 hours. The snowpack thickness reaches just over 1 m (1.1 m for Richards routine and 1.2 m for bucket routine) at its maximum in March before a fast melt out. The difference in snow thickness between the two routines is due to the wet snow compaction that relies on the LWC. The majority of the snowpack

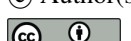



wetting occurs during the April melt. The wetting front reaches the bottom of the snowpack at approximately the same time independent of which routine was used (Fig. 5 A and C). The three rain/melt events that occur early season (23/24 Oct., 15 Nov. and 26 Nov.) pass water to the soil layers for both percolation routines. The two melt/rain events that occur after the cold period in January (7 Mar. and 6/7 Apr.) wet the snowpack's surface before it is in an isothermal state. These two events are

more pronounce and reach deeper snow layers in the simulation using Richards routine. The event on Mar 7 formed a layer with density ~300 kg m$^{-3}$(at 0.7 m to 0.9 m) when run with Richards routine, which is missing from simulations using the bucket routine (see Fig. 7 B & D).

The density evolution according to the bucket routine (Fig. 7 B) shows formation of a dense crust at the bottom of the snowpack of about 375 kg m$^{-3}$, where the Richards routine (Fig. 7 D) creates a much thicker and denser layer of 500kg m$^{-3}$. The Richards

routine makes a slightly denser pre-isothermal snowpack and a much denser snowpack during periods of enhanced water transport. The bucket routine allows easy transmission through crust layers because high density layers yield low pore volumes resulting in "small bucket size". The bucket routine does not represent crust layers well because they develop too thick and not dense enough.

## 5.2 Synthetic data set's simulation

The synthetic data set's results are presented in Fig. (8) illustrating simulations utilizing the bucket routine and Fig. (9) showing simulation using the Richards routine. Figure 8 C & 9 C are zoomed in on the second snow event with a finer plot resolution of 15min (same as T$_{Crocus}$). The percentage of pore space that is filled by water is the measure used to calculate "bucket size" which makes it an intuitive way to view water content for the bucket routine. The results of the bucket simulation show a uniform wetting front that features a stepped pattern (Fig. 8 A). The stepped pattern arises from the diurnal cycle of water input

at the surface. With the Richards routine, water percolates during the high and low phase of the diurnal radiation cycle, which results in a faster water front progression and a lack of the stepped pattern (Fig. 9 A). There is also a big variance in the percentage of pore space filled with water throughout the simulation. Effects of the diurnal cycle are visible in the Richards routine's results, in the form of percolation fronts that move down the snow layers. The percolation front passing thought the second snow event (Fig. 9 C) is delayed by layers that were deposited with different densities, which is not seen with the

bucket model.

A pattern emerges after melt water from after the second snow event reaches the bottom snow layer Fig. (9 A & C) (also seen in Fig. 7C but less pronounced). The bottom snow layer remains dry throughout the simulation, and every second snow layer remains very wet between 10-15% pore volume corresponding to a water content of 60-80 kg m$^{-3}$. Before water percolates to the bottom of the snowpack some newly wet snow layers quickly drain after the water front has percolated to deeper layers.

The bucket routine run with the synthetic forcing (Fig. 8 B), produces a thick crust layers that does not exceed a density of about 400 kg m-3. There is no delay in the water front's movement as it passes the melt freeze crust in Fig. 8B.




## 5.3 Sensitivity of pre-wetting and time step

The temperature development of the snowpack for both the Richards and bucket routines lead the wetting front. However, changing the amount of pre-wetting in the snowpack (Fig. 10) or length of $T_{Crocus}$ (Fig. 11) will change the timing of the warming and water front. A sensitivity study was performed on amount of pre-wetting using the simulated data set. The timing

of the fronts were recorded when 3 of the bottom 5 snow layers became 0°C for warming front or LWC > $\theta_{min}$ for the wetting front. The criteria of 3 out of 5 of the bottom layers reacting was chosen because it is unclear just how much of an affect the soils thickness has on the ground heat flux which can prematurely warm and wet the bottom layer. The timing between the snowpack reaching an isothermal state and the water front percolating to the bottom of the snowpack increase as pre-wetting amount increases. Fig. (10) shows a trend that the less water used to pre-wet the snowpack the longer it takes for both fronts

to propagate to deep layers. However, there is a complicated one-to-many relationship between pre-wet amount and timing of the warming and water fronts.

The time step also affects the timing of the warming and water front, which can be seen in Fig. (11). When $T_{Crocus}$ is reduced the warming and wetting front is able to percolate much faster. The alternating dry wet pattern appears before the second snow event when $T_{Crocus}$ = 60 sec. The pattern smooths out during the second snow event, probably due to rearranging of the snow

layer sizes.

## 6 Discussion

Figures 8 A and 9 A show that the Richards routine creates an isothermal snowpack and water percolation to deep layers earlier in the simulation than the bucket routine. However, the timing of the warming- and water-fronts depends on many variables. The following sections discuss some of the parameters that we have found to influence the timing and amount of water flow.

## 6.1 Liquid water content magnitude

The bucket routine has a LWC upper limit of 5% pore space. There are two wetness states that are frequented in the bucket routine, the dry state and wet state at holding capacity. A snow layer spends little time over the course of a snow season in the transition between dry and at holding capacity, which can be seen in Fig (7 C). Routines such as the compaction and grain metamorphisms were developed using the bucket routine and rely on a nearly binary water content configuration.

Using Richards routine, we obtain much higher LWCs, which constantly vary between time steps (Fig. 7 C & 9 A). Richards routine is capable of wetting snow layers to the transition between pendular and funicular regimes as defined, by Denoth (1980) where 17% of the simulated data sets snow layers that entered the percolation routine had > 10% of their pores filled with water. This study did not investigate how other routines in Crocus (e.g. the compaction routine) that are affected by the LWC are affected by high LWCs in snow layers. However, it is expected that wet snow compaction and wet grain metamorphism

are non-linear functions with feedback on LWC in the transition between pendular and funicular, since the physical distribution of water is held differently in the snow's pores with respect to snow crystals surfaces (Denoth, 1982).



## 6.2 Bottom boundary

The bottom boundary is an area of concern because it feeds the soil water percolation routine with meltwater and differences in pore structure over the snow soil interface will often create at textural barrier, which can lead to pooling or accelerated flow. Figure 7 C & 9 A , show the bottom snow layer remaining drier than then second bottom snow layers after the snowpack is

fully wet. The hydraulic conductivity of the bottom soil layer and the suction of the soil layer are imported to Crocus from the soil routine. These values are held fixed for the time step $T_{Crocus}$. The hydraulic conductivity of the bottom interface is the arithmetic mean (harmonic mean can be used) of the bottom snow layer and the top soil layer's thickness and hydraulic conductivity. Snow layers are able to move water between them on the internal time step (t), which can be as small as a fraction of a second. Snow layers suction and conductivity are altered on the internal time step (t). However, the soils suction and

conductivity remains constant until $\Sigma t=T_{Crocus}$ witch means the snow-soil interface has a less dynamic conductivity and suction because half the values making up the $K_{snow/soil}$ are constant on $T_{crocus}$. One way to force the bottom interface to be more dynamic is to reduce $T_{Crocus}$ (Sect. 5.3). Ideally, the snowpack and soil column would be solved as one continuous column (Wever et al., 2014). However, the snowpack is semi-implicitly coupled to the soil percolation routine on $T_{Crocus}$. Unfortunately, with the ISBA model the soil and snow routines are not coupled on the variable time step t. To solve the soil and snow as one

continuous column of unsaturated porous media would take major reorganizing of the SURFEX model and was not feasible during this study, but should be considered in the future.

During periods of high flux between the snow and soil, the convergence criteria will not be reached with a reasonable time step (t). It was found that increasing the size of the top soil layer allowed the convergence during high flux periods. However, increasing the size of the soil reduces the reliability of snow, soil thermodynamics routine, but that is beyond the scope of this

study. The size of the top soil layer was set to a value which allowed the simulation to converge in a reasonable amount of time. For the Filefjell simulation the top soil layer was 1.0 m and for the simulated data sets simulation the soil layer was 0.3 m.

## 6.3 Coupling

$T_{Crocus}$ dictates the degree of coupling between SURFEX routines, including the coupling between the ISBA and Crocus and

between routines in Crocus. The water percolations process is coupled to the soil percolation process and the energy balance of the snowpack, in Crocus these routines are run sequentially and are coupled via $T_{Crocus}$. The energy balance is made up of many processes that are expressed through many of the routines in Crocus (see Fig. 1). The temperature of the snow layers is altered after the percolation routines (for both Richards and bucket) due to latent heat release, if refreezing occurs. Although, a sensitivity study on the link between temperature of snow layers and percolation would be relevant such a study was not

possible because of other feedbacks in the model.

The histograms in Fig. (2 & 3) show the distribution of snow layer's saturation when they entered the Richards routine. The water content is often very low with 69.5% of the snow layers under 5% LWC. At low saturations, the hydraulic conductivity



function and the retention curve are very sensitive to changes in density, grain size and saturation. It is important to note that a 1% change in saturation when very dry can affect both the hydraulic conductivity and the water retention curves by orders of magnitude. Figure 11 shows the simulated forcing when run with $T_{Crocus}$ = 900 sec and $T_{Crocus}$ = 60 seconds. Water moves through the snowpack earlier when $T_{Crocus}$ = 60 seconds because faster feedback between the percolation routine and routines

alters layer density and layer grain size.

Similar to snow layers, a small change in saturation of soil layers should affect the hydraulic conductivity and the suction of the bottom boundary. However, the top soil layer is only coupled to the snowpack via $T_{Crocus}$. Problems occur when too much water is passed from snowpack to soil in one time step (also discussed in Sec. 6.2). If $T_{Crocus}$ is large and the top soil layer is thin, the amount of water transfer from the snow into the soil would drastically change the conductivity and suction at the soil-

snow interface in a time span much smaller than $T_{Crocus}$ and the routine does not converge. Reducing $T_{Crocus}$ is a strategy that will better couple the soil layers and snowpack. This strategy is not feasible however, since the Richards routine is sensitive to the duration of the time step ($T_{Crocus}$), the approach of increasing the soil layers size is preferred.

### 6.4 Pre wetting amount

Water percolation and temperature development are sensitive to large values of $\theta_{min}$ ($>10^{-6}$), which is the amount of pre-wetting

used. The pre-wetting is needed because the water retention curve approaches infinity and the hydraulic conductivity function approaches negative infinity when $\theta$ approaches 0 (see Fig. 2 & 3). Figure 10 shows that the amount of pre-wetting used strongly affects the timing of when the snowpack reaches an isothermal state, and but does not have a large effect on the water front movement. With $\theta_{min} \geq 25\times10^{-6}$ ($m^3$ $m^{-3}$) used for the pre-wetting amount, the snowpack reaches an isothermal state quickly after the top layers of the snow have been introduced to a sufficient amount of melt water (day 39), such that the

criteria to enter the percolation routine has been met. The water front does not show a pattern concerning $\theta_{min}$. The water front reached the lower snow layers on day 58 or 59 regardless of the $\theta_{min}$ limit. Latent heat released in deeper snow layers from the freezing process are an effective way of transporting energy through the snowpack. It is unlikely that there are long periods between the warming front and the wetting fronts movement in the snowpack. $\theta_{min} = 10^{-6}$ was chosen as default because the two fronts started to reach deep snow layers together, where a larger value of $\theta_{min}$ resulted in a warming front leading the

wetting front by more than a day. The time between the warming front and water front reaching the lower layers of the snow pack constantly decreased as $\theta_{min}$ decreases. The bucket routines results show that the warming and wetting fronts move together down the snowpack.

### 6.5 Conductivity through crusts

The Richards routine is able to produce crusts layers from melt water that the bucket routine cannot reproduce. However, the

thickness and density of crust layers are dependent on where the percolation routine moves water since refrozen liquid water is often the cause of crusts at the surface and inside the snowpack. The water retention curve and hydraulic conductivity functions are not designed for use on crust layers. Furthermore, Crocus' snow metamorphism routine B92 does not work well



for crust layers, because dense crusts do not have individual grains, but rather a solid ice layer with bubbles. The C13 routine uses optical diameter, which is used as an approximation for visual grain size in the Richards routine. The assumption that optical diameter is a sound approximation for visual grain size is questionable, especially for crust layers. Nevertheless the metamorphism routine calculates a grain size for all layers including crusts (see Sect. 6.6). Figure (9B) shows that the crust

layer was able to develop into a thinner and denser crust compared with the bucket routine (Fig. 8 B). Since there is no literature available on water retention and hydraulic conductivity of crusts layers, crusts are treated like normal snow layers in both percolation routines. The Richards equation is solved in one dimension normal to the snow surface. However, dye tracer experiments have shown that water transport though snow is a two- (maybe three-) dimensional process (Williams et al., 2010). When the wetting front reaches a barrier such as a crust or capillary differences, the underlying snow layer probably will

develop flow channels where a one dimension model does not suffice. Averaging the hydraulic conductivity between a layers barrier suction with the flow channel suction is one way to express a two dimensional process in one dimension. Hydraulic conductivity has been determined by Calonne et al., (2012) on small snow samples therefore it is likely that the measurements do not represent conductivity at a higher spatial scale because preferential paths have a spatial pattern on a 5 to 7 m scale (Williams et al 1999).

### 6.6 Grain metamorphism routine

The C13 routine was used for snow metamorphism in the presented simulations, which uses optical diameter.

The hydraulic conductivity function from Calonne et al., (2012) utilizes the optical grain diameter and snow density. The hydraulic conductivity function pairs well with the C13 routine, because they both use optical grain diameter. However, the water retention curve (Yamaguchi et al., 2012) was based on a visual grain size measurements and the use of optical grain

diameter may hinder the performance of the water retention curve.

The pore shape and structure is important for the retention curve and the hydraulic conductivity. Small pores inside the snow layer create suction via capillary rise, and water travels through the voids in the snowpack. Both parameterizations could benefit from a better description of the pores structure. Density and grain size is not enough to describe the pore structure of a snow sample. We use optical diameter for parametrization but in nature, optical properties do not influence the snows

hydrological processes. For the retention curve and the conductivity functions to be used at low saturations, pore shape and structure should be considered. Introducing a routine in Crocus that calculates pore shape could be beneficial for not only the water percolation routine but also the thermal conductivity (Riche and Schneebeli, 2013; Sturm et al., 1997). This would also require new parameterizations including a pore structure variable with the grain size and density.

### 6.7 Water retention curve

The Yamaguchi et al. (2012) water retention curve has been applied to melt forms crystal type and reported poor performance with rounded grains. It is expected that the retention curve does not represent precipitation particles, decomposing and fragmented precipitation particles, faceted crystals and depth hoar well. However, when water is present in a snow layer snow



grain metamorphism will transform all snow crystals into MF (Colbeck, 1976; Shimizu, 1970). This negative feedback on snow grain type could mean the melt forms are the only crystals that need a modeled water retention curve, but this claim needs validation.

The residual water content is one of four parameters needed to define the water retention curve. The residual water content

represents how dry a layer can get with maximum suction applied. A constant $\theta_r$ was used for different grain types sizes and densities but results from Adachi et al. (2012) suggest that $\theta_r$ varies with grain size.

Hysteresis in the water retention curve stems from odd shape pores where, pores can hold different amounts of water at the same hydraulic head depending on the initial LWC. In most cases, snow will go from a dry state (pore space filled completely with air) and become wet. The Yamaguchi parameters for the VG model are derived from a drainage experiment, where the

pore space was completely filled with water and allowed to drain. Adachi et al. (2012) showed that the shape of the water retention curve is affected more in fine textured grain then coarse grains.

Parametrization of the water retention curve based on a wider range of snow grain types and sizes are needed to represent natural snow pack conditions. For fine textured snow, parameter sets derived from wetting experiments would be beneficial, as the snow usually starts from a dry state.

**7 Conclusion**

We added a new water percolation routine to the snow pack model Crocus as an alternative to the empirical bucket percolation routine. The bucket model keeps snow layers in the pendular wetness regime. The Richards routine reaches LWCs much higher than the bucket routine, with LWC >17% for many snow layers, which lies in the funicular regime for all snow types. However, 83% of snow layers were initialized in the Richards routine with a LWC <10%. With small changes in LWC (>1%) at low

saturation the suction and hydraulic conductivity can change by orders of magnitude. The parameterization used to define the suction and the hydraulic conductivity is heavily reliant on the snow grain size and the density of snow layers. The wet snow metamorphism and wet snow compaction routines were implemented when the bucket model was the only option for water transport. There is a physical difference in the distribution of water inside the pores between pendular and funicular regimes. This difference is not accounted for in the snow compaction rate or wet snow metamorphism rates, which leaves the feedback

to the new routine open to question.

New parameterizations for the hydraulic conductivity and the water retention curve that do not contain grain size are needed for dense crust layers. The Richards routine in the current state treats crusts as a normal snow layer where grain size calculations are erroneous. The parameterization for the water retention curve was based on a small domain of densities, grain sizes and snow types, the domain for this parameter set should be expanded for applicability in snow grains other than MF. New

parameter sets should be based on wetting experiments for small grain sizes as hysteresis affects the Van Genuchten parameters ($\theta_r$, $\alpha$, n).



The arithmetic mean was used to define the conductivity of interfaces between snow layers. By doing this, the stratified nature of the snow is smoothed. Different expressions for the conductivity at interfaces, yielded subpar results. The behavior of the hydraulic conductivity at interfaces, is not well understood/expressed in current literature, but vital for calculating water transport through crust layers.

At the current state of the Crocus model, major structural changes are needed in order to couple the soil and snow percolation routines. We identified the soil-snow interface as the source of numerical instabilities, when the top soil layer is small (< 0.3 m). This study did not investigate the effects the Richards routine has on the ISBA soil percolation routine, or the effects of a large top soil layer.

Because of the number of areas of concern in the validity of the Richards routine, we did not attempt a validation experiment.

**8 Code and data availability**

The SURFEX/CrocusV7 with Richards routine can be download via SVN by following instructions at https://opensource.cnrm-game-meteo.fr/projects/snowtools/wiki/Install_SURFEX, and checking out the branch "damboise_dev". A SVN account can be requested by following instructions given in link: https://opensource.cnrm-game-meteo.fr/projects/snowtools/wiki/Procedure_for_new_users.

The Filefjell and simulated data set with the namelist needed for running the simulations are provided at, www.norstore.no (internal identifier AA4769C6-5277-4523-A3F0-7155B2DC39EC, & 2CEC0984-50A1-4A2A-B6C9-A83815EA50E8). Internal identifiers are used because the data set is in preparation, a doi reference will be provided when data set is published.

**Appendix A - Picard iteration/convergence**

A Picard iteration is used deal with the non-linear nature of the system of equations. There are three convergence tests that
need to be satisfied to calculate the next internal time step (t). The three convergence criteria are on pressure head differences, volumetric water content differences and the mass balance (of individual snow layers) differences between two iterations must be below $10^{-4}$. Hence, there must be at least two iterations performed in order to test for convergence. If the current iterations deviation of pressure, volumetric water content and mass balance is smaller than the error-threshold that has been set then the convergence criteria is met and the model proceeds to the next time step.

A set of rules are used to regulate the variable time step based on the number of iterations needed for convergence. A maximum number of iterations has been set to 15. If the calculations do not converge by the 15th iteration then the time step is reduced and calculations are preformed again, until the calculations converge within 15 time steps. The variable time step can range in time from about 400 seconds to fractions of a second with a lower limit of 1E-10 seconds. Smaller time steps with in the Richards routine results in the model taking too long to run or convergence is not achieved. If convergence is reached in 4-7



iterations the time step is kept constant. Converging in less than 4 or between 8-15 iterations results in a longer or shorter time step respectively (Eq. A1).

$$t^{i+1} = \begin{cases} t^i \cdot 1.5^{BS}, & I < 4 \\ t^i, & 4 \leq I < 7 \\ t^i \cdot 0.5^{SS}, & 7 \leq I < 15 \\ back\ step, & 15 < I \end{cases}$$

Where I is the number of iterations and "back step" stops the calculation before convergence and reduces the $t^i$. BS and SS are initialized at 1 and is increased by 1 for the previous "bigger steps" or "smaller steps". BS and SS are reinitialized if the time step remains the same size ($4 \leq I < 7$).

**Appendix B – Alternative parameter sets**

The following sections describe alternative parameter sets for hydraulic conductivity (B.1) and water retention curve (B.2, B.3) that we implemented in the Richards routine, but not presented in this paper.

**B.1 Shimizu 1970**

Shimizu developed a statistical model for $k_{sat}$ from the density and grain size of fine grained compact snow, Eq. (B1). Shimizu suggests that other snow grain types and densities may give different coefficients than those found in Eq. (17). The parameter set has been applied to lower density snow that found capillary pressure is related to pore structure (Jordan et al 1999). Equation 17 relates snow density and grain size to conductivity at saturation ($k_{sat}$).

$$k_{sat} = 0.077 \cdot \left(\frac{D}{2}\right)^2 \cdot exp(-0.078 \cdot \rho_{snow}) \cdot \left(\frac{G \cdot \rho_{water}}{\mu_{water}}\right) \tag{B1}$$

**B.2 Yamaguchi et al 2010**

This parameter set was the prior work of the parameter set used in this study. Two major differences (when compared with Yamaguchi et al. 2012) for this parameter set are $\alpha$ and n are functions of grain diameter (D) (density not include) and the density range tested in this study is small 545-553 kg m[-3]. While the parameterization for the $\theta_r$ and $\theta_s$ are the same as in Yamaguchi 2012.

$$\alpha = 7.3 \cdot (D \cdot 1000) + 1.90 \tag{B2}$$
$$n = 15.68 \cdot exp(D \cdot 1000) \cdot (-0.46) + 1.00$$

**B.3 Dannen & Nieber 2009**

Dannen and Nieber derived these relations based on measurements of liquid water content and water pressure from Marsh (1991). They assume snow crystals between 1mm and 0.1mm. However, descriptions of the measurements are not given in



detail. Dannen and Niebers study focuses on a framework of a coupled temperature and liquid water routine that does not focus on deriving the Van Genuchten fit parameters in a suitable way for different snow types, as all crystals are assumed spheres.

$$\alpha = 30 \cdot (D \cdot 1000) + 12.00 \qquad\qquad (B3)$$

$$n = 0.8 \cdot (D \cdot 1000) + 3.00$$

$\theta_r = 0.05$

$\theta_s = \text{Porosity}$

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



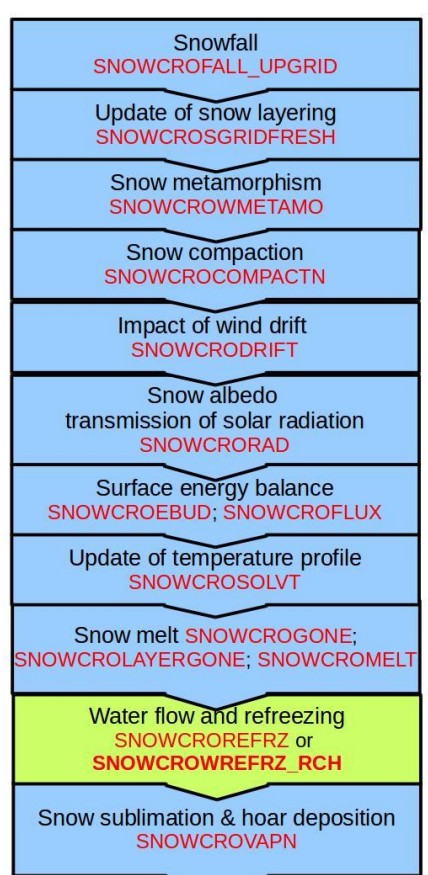

**Figure 1: Routines in the Crocus snowpack model with the water percolation routines highlighted in green.**





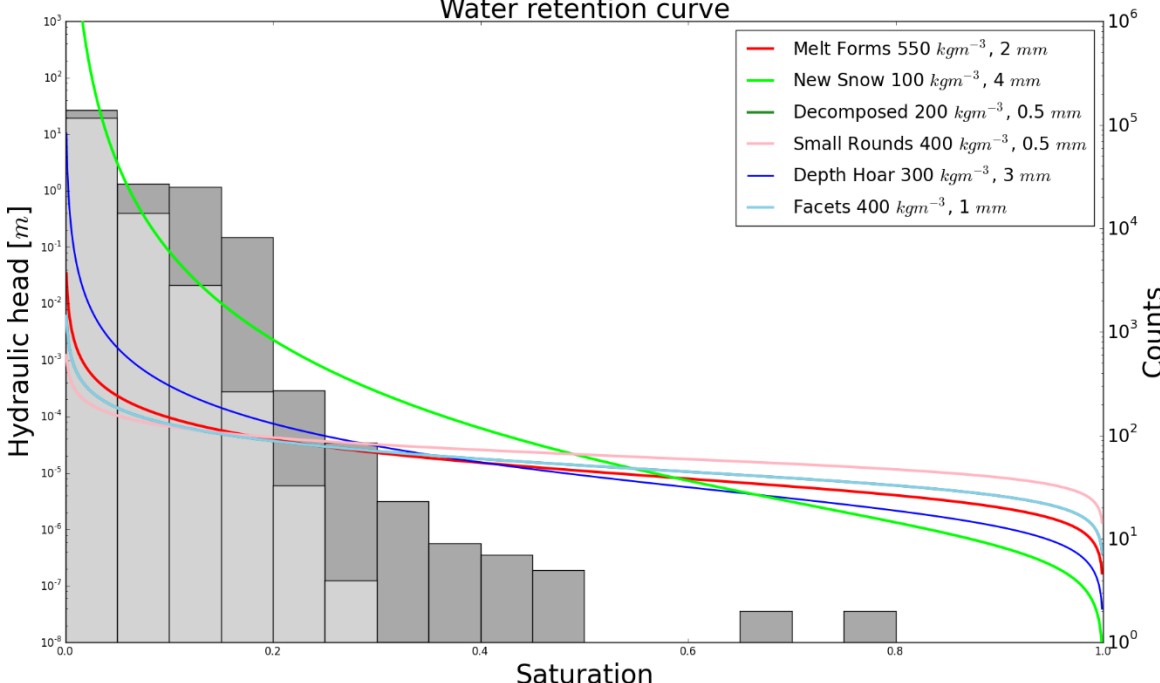

**Figure 2: Water retention curve using Yamaguchi 2012 in the Van Genuchten (1980) parameterization. Typical values for density and grain sizes of different snow crystals/grains were chosen to show the water retention curve for the spectrum of different snow layers it is applied to in the Richards routine. The density and grain size values were chosen from each crystal type from within a reasonable range that may be found in nature. Background shows a histogram of the simulated data sets saturation with T$_{Crocus}$= 900 s time step before entering the Richards routine. Dark grey bars show the full 90 days. Light grey bars show the first 75 days prior to the period of high flux between the snow and soil**





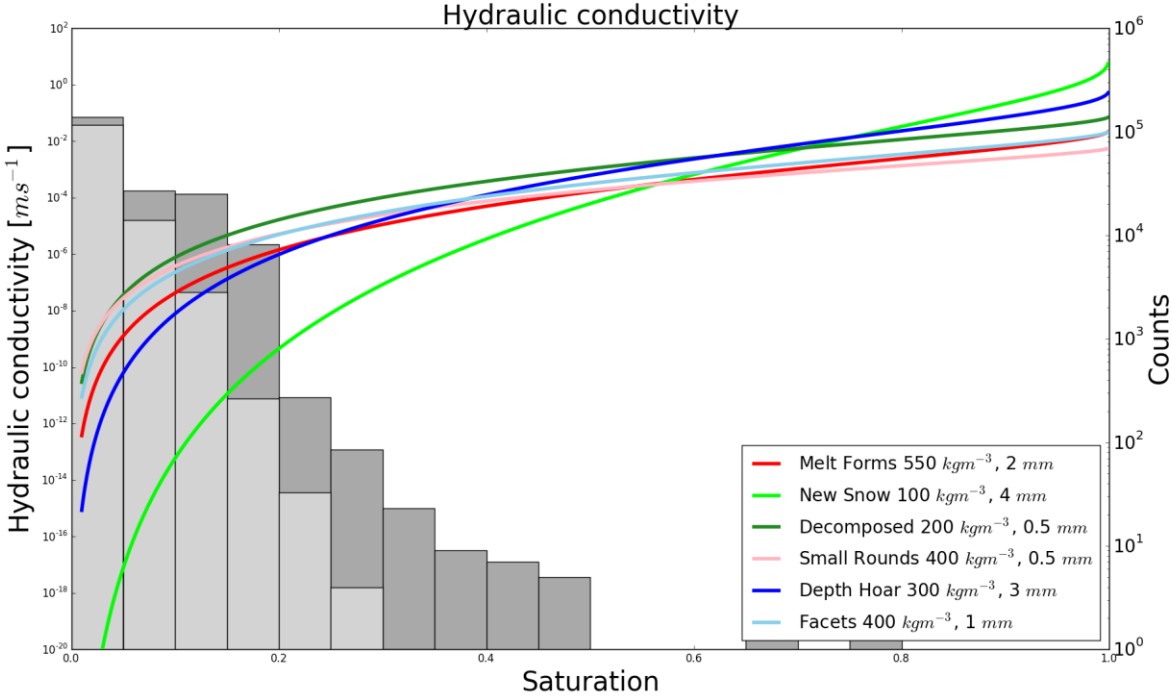

**Figure 3: Hydraulic conductivity curve from Calonne et al. (2012) using Yamaguchi et al. (2012) parameters. Typical values for density and grain sizes of different snow crystals/grains were chosen to show the hydraulic conductivity for the spectrum of different snow layers it is applied to in the Richards routine. The density and grain size values were chosen from each crystal type from within a reasonable range that may be found in nature. Background shows a histogram of the simulated data sets saturation with $T_{Crocus}$= 900 s time step before entering the Richards routine. Dark grey bars show the full 90 days. Light grey bars show the first 75 days prior to the period of high flux between the snow and soil**



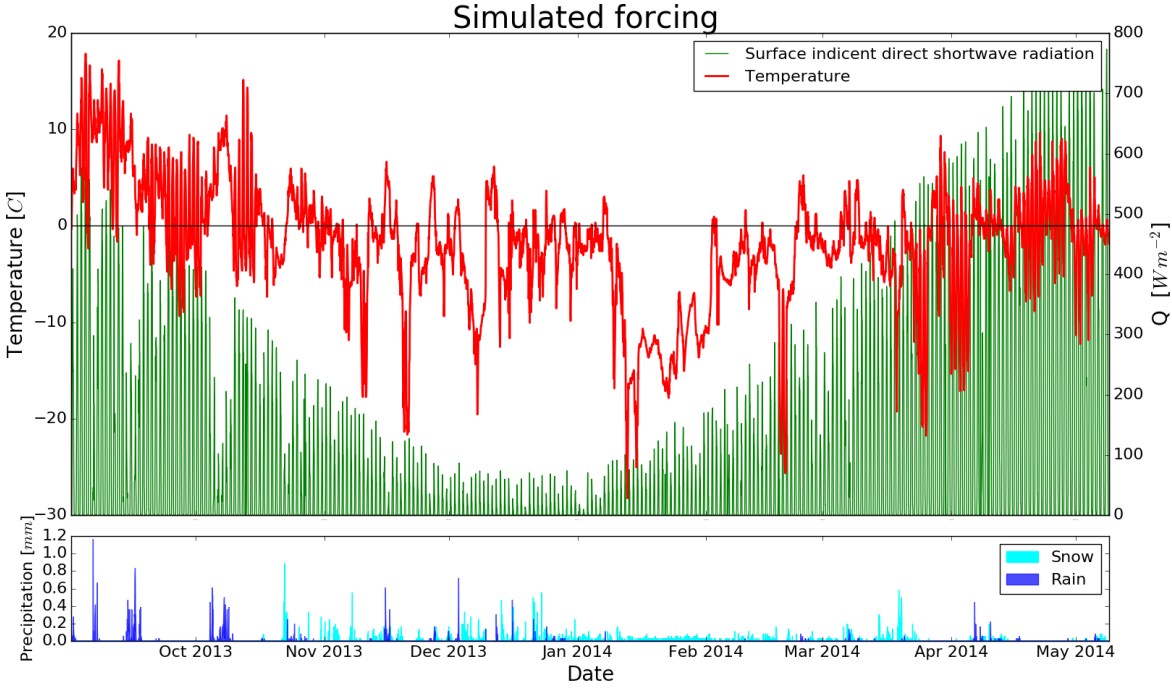

**Figure 4: Forcing data from an automatic weather station located in Filefjell, Norway for the period 01.09.2013 to 31.05.2014.**



**Figure 5: Snow profile from ~ 3km away from the Filefjell field site on a 24° SE facing slope. (Solemsli, 2014)**



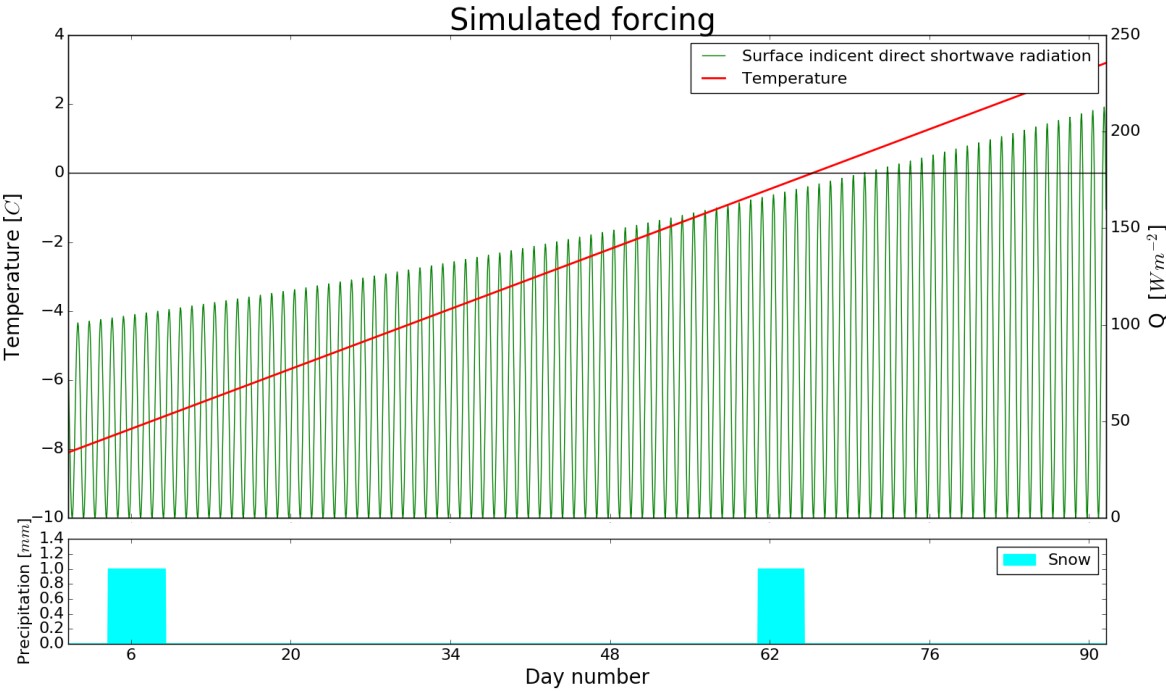

**Figure 6: Simulated forcing data. Temperature (red) and net radiation (green) both linearly increasing creating low melt early in the simulation and high melt at the end of the data set.**





**Figure 7: Crocus output from Filefjell, Norway, where top plots use the bucket routine and bottom plots use the Richards routine. Plots A and C show distribution of liquid water, and plots B and D show the density distribution. T_crocus=900 s (15 min) was the time step duration. For the Richards simulation (C + D) the bucket routine was used for melt out when snowpack was less than 0.1m (default is 0.05m).**



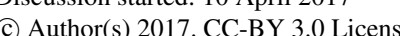

**Figure 8:** Crocus output for Neverland forcing using the bucket routine with T$_{Crocus}$ = 900 s (15 min), plotted every 3 hours, except plot C, which is plotted every 15 minutes. A) liquid water amount, B) snow layer density, C) % pore volume filled with water zoomed in on the second snow event, and D) temperature development of the snowpack.





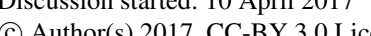

**Figure 9: Crocus output for Neverland forcing using the Richards routine with $T_{Crocus} = 900$ s (15 min), plotted every 3 hours, except plot C, which is plotted every 15 minutes. With $\theta_{min} = 10^{-7}$. A) liquid water amount, B) snow layer density, C) % pore volume filled with water zoomed in on the second snow event, and D) temperature development of the snowpack.**

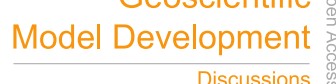



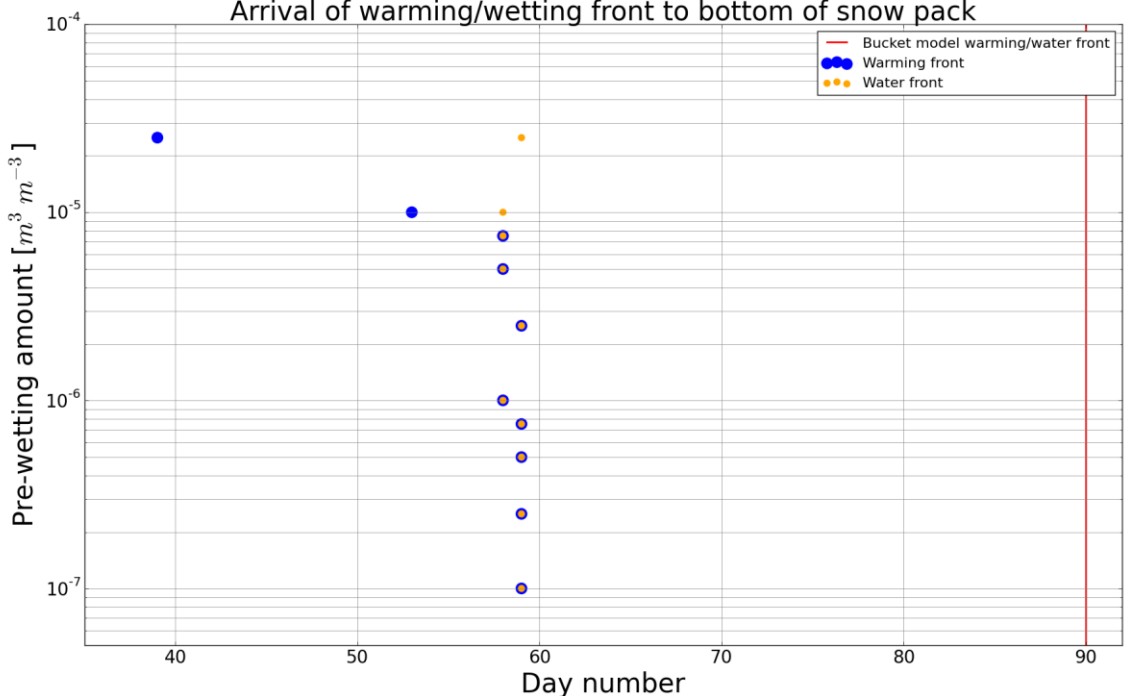

**Figure 10: Timing of warming/water front to reach bottom of the snowpack with different amounts of pre-wetting. T$_{Crocus}$= 900 s was used to make this figure. Blue marks represent the timing the snowpack reached an isothermal state, where orange marks show when the water front reached the 3rd lowest snow layer. Red line is when the bucket routine's warming and wetting fronts reached the bottom of the snowpack.**





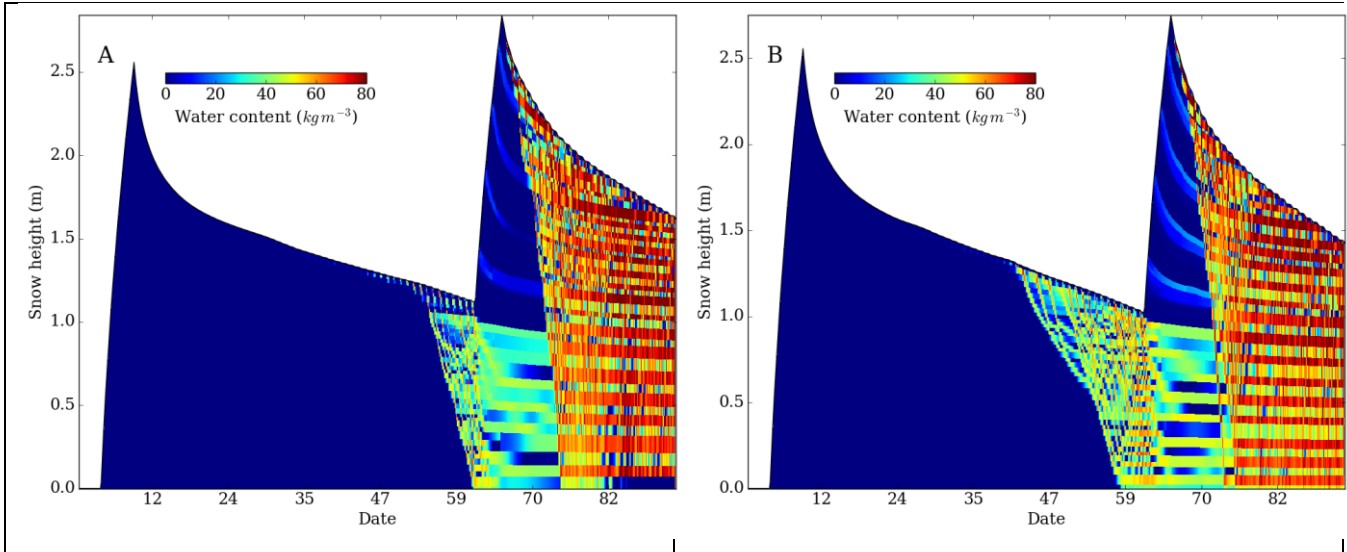

**Figure 11: Water percolation in the simulated data set scenario with different time steps: A) T$_{Crocus}$= 900 s (15 min), B) T$_{Crocus}$= 60 s both plots use θ$_{min}$ = 10$^{-6}$.**

