# Peer review of "Implementation of a physically based water percolation routine in the Crocus/SURFEX (V7.3) snowpack model"

_Geoscientific Model Development, 2017_

## Referee Comment (RC1) · N. Wever (Referee) · 2 May 2017

**1   General comments**

The manuscript by d'Amboise et al. discusses the implementation of a solver for Richards equation into the detailed, multi-layer Crocus snow model. This equation describes water flow in porous media, such as snow, and previous studies have already shown that snowpack models can benefit from implementing this equation. It generally seems to provide a better representation of liquid water content (LWC) distributions and snowpack runoff behaviour. In that sense, the study represents an important step for the Crocus snow model. Also the study can be considered an important independent verification of the results achieved with the SNOWPACK model, where the solver for Richards equation has been found to considerably improve the description of liquid water flow in snow in several aspects. I found the manuscript well written and pleasant to read, but there are also some language and grammar issues (see technical corrections below). I value and appreciate the effort undertaken by the Crocus team. Based on my experience, I know that introducing new routines to a snowpack model with such a big impact as a new water percolation routine is a serious and difficult piece of work.

My first main concern with the study presented by the authors is that the results presented here are not convincingly showing that the model behaves numerically stable. Distributions of liquid water content look different from distributions achieved with the SNOWPACK model. The absence of a comparison with field data of profiles of liquid water content or snowpack runoff makes it impossible to judge the validity of these results. I will do a few suggestions for additional verification of the numerical scheme, which I hope will provide convincing evidence that the model behaves numerically stable. My second main concern is that the general message of the manuscript is not clear and very open and may potentially confuse readers (see below).

**2 Specific comments**

I have the following remarks related to the numerical scheme:

1. Especially the alternating wet and dry snow layers shown in Fig. 9 and 11, and discussed in p10,l27, are very suspicious. It looks like a numerically oscillating solution. If it is a true LWC distribution, it is recommended to have a higher vertical resolution in the simulations (i.e., more snow layers) in order to better represent the strong gradients between the wet and dry snow layers. But the simulated values of 10%-15% seem unrealistic. Such high values may occur occasionally above capillary barriers, as shown in the work by *Avanzi et al.* (2016), but I'm not convinced that it should happen so regularly in the simulations shown in Fig. 9 and 11. Particularly because the artificial large snow falls create a very homogeneous stratification, such that ponding is not expected to occur. So actually I wonder if this is not a representation of the fact that the Crank-Nicolson scheme can be prone to spurious oscillations? As far as I know, Crank-Nicolson schemes are generally considered globally stable, but irregular initial conditions may lead to oscillatory behaviour. The current simulations are done with only the optimal time step for Richards equation. But are the model results sensitive to the time step inside the Richards solver? If the model is forced to run with much smaller time steps, are the results different? Or if the model is run with higher grid resolution, or by switching of remeshing, are the results different? As far as I know, the oscillations from the Crank-Nicolson scheme can be reduced by smaller time steps and/or higher grid resolution. Also stability criteria for Crank Nicolson schemes exists, which could be discussed by the authors. Note that I also have a suggestion to initialize the model more stable, see point 6 below, which may also improve numerical stability. Maybe if possible also provide additional motivation for the choice of a semi-implicit scheme instead of a fully implicit one. Is the discretization (Crank-Nicolson + Picard iteration for Richards equation) used here newly developed in this work or has it been applied before? If so, please add the references.

2. The mass balance is verified in the Picard scheme with a threshold of $10^{-4}$. The authors should add units here, but for now I assume it is the mass balance error in $m^3/m^3$ or $kg/m^2$. I think that this value is set too large to judge mass conserving behaviour of the model. The minimum time step in the solver is $10^{-10}$ s (p16,l28). If the solver has a mass balance error of $10^{-4}$ with a time step of $10^{-10}$ s, this implies a mass balance error of $1^6$ $m^3/m^{-3}/s$ or $1^{10}$ $kg/m^2/s$, or $1^6$ $kg/m^2/s$, depending on the units. But this is a potentially large mass balance error! This

means that if bugs in the numerical scheme or in the implementation of boundary conditions exist, the solver can "cheat" upon the mass balance check by choosing small time steps. Note that in the current version of SNOWPACK, this is also possible. During development of the solver we were particularly paying attention to the smallest time step in combination with the mass balance check. However, we relaxed the condition, by setting a low minimum time step, also allowing the solver to "cheat" the mass balance check. The motivation is to have a more robust solver for end-users. For this review, I analysed the time step distribution for running 15 years of Weissfluhjoch simulations using Richards equation with SNOWPACK, and the smallest time step during this period is about $2 \times 10^{-5}$ s. The maximum allowed mass balance error in SNOWPACK is $1 \times 10^{-10}$ kg/m$^2$ for the entire model domain. The combination of smallest time step found in this simulation setup, together with the mass balance criterion gives $5 \times 10^{-6}$ kg/m$^2$/s. As can be seen in the Fig. 1 below, these small time steps happen very seldom for 15 years of simulations.

3. A check of the second norm of the deficit vector could help to verify the correct implementation of the matrix inversion to solve the equation. Given: A . b = x, where x is the new solution of the pressure head, then the deficit vector is defined as d = A . b - x, where d is the deficit vector. It should hold that the second norm of the vector, in case of correct implementation, is (close to) 0. Note that in an attempt to optimize the execution time of the SNOWPACK model, we removed the deficit norm check from the code, after using it first to successfully verify a correct implementation of the solver using this check.

4. An overall report for the Crocus model as a whole of the mass balance may also be necessary to verify a correct implementation: Delta SWE = evaporation/condensation + sublimation/deposition + snow/rain fall + runoff?

5. With the numerical scheme for Richards equation in the SNOWPACK model, we

found that an improved stability was achieved by initializing dry snow layers (the authors call it "prewetting") based on pressure head instead of LWC. The chosen form of Richards equation uses gradients in pressure head, thus it may be better to reduce gradients in pressure head when initializing the dry snow layers. Therefore, we used the procedure to determine for the whole domain the lowest pressure head in a layer corresponding to a prescribed minimum LWC. This value of the pressure head was then used to initialize dry snow layers, such that only gradients due to gravity are present. It ensures that no snow layer is initialized with a LWC above the prescribed minimum value, while at the same time starting the simulation with a numerically stable pressure head distribution.

6. I noticed in the source code that C (dtheta/dh) is limited to $-1^{-15}$. For what reason? C is supposed to be the exact derivative of the water retention curve. An artificial cut-off seems unnecessary and may introduce mass balance errors?

7. Section 6.2 and p16,l5-6: why not implement a free-drainage boundary condition at the bottom of the snowpack, instead of all the trouble it seems to give to use the SURFEX upper soil layer? We recently modified the SNOWPACK model such that it can run Richards equation without soil layers, and using a free drainage boundary condition seemed to work well. In SNOWPACK, we implemented free-drainage by setting the flux at the bottom of the lowest snow element similar to the flux at the top of the element, while only allowing downward flux (otherwise setting flux to 0). In case of only one snow element, we set the flux equal to the hydraulic conductivity in this element.

I have the following remarks regarding the manuscript itself:

1. The message of the manuscript is ultimately very unclear and open. The authors apparently don't trust the new water percolation scheme enough to use it for validation (p16,l9). That basically indicates to readers that this publication is not

intended to encourage users of the Crocus model to use the new percolation scheme. But what is then the main message of the manuscript? What are the next steps to improve the trust in the validity of the new routine? Are there any further developments needed or planned? The authors should also provide clear instructions of how to repeat the experiments. I was able to download the source code, but did not find any manual or readme to compile nor did it seem to contain the necessary files to run the test cases.

2. I don't agree with the last sentence of the abstract. First, the absence of validation limits the value of any comment about applicability, but basically the same uncertainties in water retention curves for different snow types and for high density crusts, and also many of the feedback mechanisms are present in the SNOWPACK model too. Nevertheless, we have now demonstrated several times that solving Richards Equation is having usefull applications for the SNOWPACK model, in spite of all the uncertainties. For example for assessing wet snow stability (*Wever et al.*, 2016a; *Vera Valero et al.*, 2016) as well as in a detailed analysis of rain-on-snow events (*Würzer et al.*, 2017) and for reproducing ice layers (*Wever et al.*, 2016b). As shown in Table 1 in *Wever et al.* (2015), different water retention curves or different methods to determine the hydraulic conductivity at the interface nodes (arithmetic vs geometric) have limited influence on the statistics for runoff, whereas the statistics clearly improve over the bucket scheme. Therefore, I don't agree with the statement of "limited applicability" with the reasons provided in the rest of the sentence.

3. The discussion section is nicely written and provides an interesting introspective discussion about the uncertainties and potential feedback mechanisms in water percolation modelling. Note that, however, many of the feedbacks are hypothesized or based on results of other studies. The manuscript itself does not present material supporting or quantifying the strength of those feedbacks (no validation or sensitivity study). It would be good if the discussion could be made stronger.

For example, the authors may want to discuss how water retention curves for crusts potentially look like, and how strong this influences LWC distributions or snowpack runoff?

4. p4, section 2.1: Maybe explicitly state that the working of the bucket scheme in Crocus is very similar to the SNOWPACK model. p4,l9: Is the bucket size always fixed to 5% of pore space? The sentence following this sentence is a bit confusing, as if there are additional constraints. Should the sentence "For Crocus, ..." not better read "This makes the holding capacity proportional to the density of the snow layer, *but* independent of snow grain type or surrounding environment." It it also not clear what is meant by "surrounding environment" and how it could potentially influence the holding capacity?

5. p8,l14: please provide a bit more detail on how the amount of evaporation is determined. Atmospheric forcing only provides the latent heat flux, which needs to be partitioned in evaporation and sublimation. How is Crocus doing it? Note that a reason for numerical problems with Richards equation can be when the prescribe evaporation flux exceeds the available water. For this reason SNOWPACK employs a system where the evaporation cannot exceed the amount of water available in the upper element plus the amount of water that can be advected from below given the hydraulic conductivity there. Similar for influx, although typically unrealistic large rainfall rates ($>>$ 200 mm/hr) are necessary to exceed the absorption limit in snow of liquid water. In reality melt ponds form in snow only when liquid water cannot leave the snowpack below, not because the water input rate exceeds the snow absorption capacity.

6. p13,l20: "such that the criteria to enter the percolation routine has been met." This is very confusing at it is for the first time mentioned that there are criteria whether or not to enter the percolation routine. Which criteria are meant here?

7. p14,l13: Many examples can be found to show that preferential flow has a much

smaller typical spatial scale. See for example Fig. 2 in Techel and Pielmeier (2011), or Fig. 1a in *Würzer et al.* (2017). Many other examples can be found in literature.

8. p15,l2-3: "this claim needs validation": I think it depends on the application. During the first wetting, grain shape will probably play a very important role. New snow getting wet probably retains much more liquid water initially than the water retention curve developed for melt forms will provide. For wet snow avalanche prediction, the first-wetting is often considered of crucial importance and I think improvements in the description of water flow in new snow and faceted snow (generally less shear strength) are required. On the other hand, for many hydrological applications, often the runoff behaviour during a melt season is important, for which the assumption of melt forms is justified.

9. p15,l19: This is a bit confusing wording, as principally, I would say that snow layers are initialized with the "pre-wetting" amount. But here, it is probably meant initialization when the routine is being called during a Crocus time-step.

10. p15,l26-28: Although I agree with the statement, it cannot be considered a conclusion of *this* work. It has not been demonstrated that the water flux over the crust is over- or underestimated (no validation done), neither has it been shown that the simulations are sensitive to the hydraulic parameters for crusts (no sensitivity study done).

11. Fig. 5: This figure is only mentioned once in the manuscript, and is not discussed at all. Please discuss the agreement between model and observation, or remove the snow profile.

12. I think the manuscript should not only show results for LWC distributions inside the snowpack, but also snowpack runoff.

**3 Technical corrections / minor comments:**

- General: note that *Calonne et al.* (2012) determined permeability, which can be related to conductivity. But in principle they did not publish conductivity experiments.

- General: please mention somewhere the CPU time needed to run the simulations, compared to the bucket scheme.

- General: there is a change of tense sometimes, compare for example section 4.3 with 3.4.

- General: sometimes "Figure" and sometimes "Fig." is used to refer to figures.

- Abstract: "this routine is based on". Why the wording "based on"? I would write here: "this routine solves Richards equation".

- p2,l2-3: note that simulations using Richards Equation have already been used for the assessment of wet snow avalanche activity (*Wever et al.*, 2016a) and for determining the initial conditions for wet snow avalanche dynamics simulations (*Vera Valero et al.*, 2016). Given that the authors discuss this topic in particular, they may consider citing these studies here.

- p2,l16: This would not be the way I would explain the precondition for flow fingering, but I also have not the evidence to object against it. Maybe verify with *DiCarlo* (2013)? I think this is a more up-to-date citation that may be cited here aswell.

- p3,l3: "Greenland Ice Sheet" (capitalized)

- p3,l8: "ice crusts": in line with the International Classification (*Fierz et al.*, 2009), this should be "layers". The mentioned study concerned more with ice layers than

with ice crusts. Similar p3,l4: lenses are discontinuous ice layers. In this case, I think it should be layers rather than lenses.

- p6,l19: "kr" should be $k_r$.

- p7,l17: Here and elsewhere: I prefer "conductivity of snow"

- p7,eq 13: I assume the second equation should read $\Delta z_{bot}$

- p7,l14: I assume the reference should point to Eq. 13 instead of 11.

- p7,l24: citation style of Celia et al. is wrong (without parenthesis)

- p8,l3: "computation step" is a vague term. I think this refers to the iteration level k+1? Maybe write: "the pressure head h at iteration level k+1.".

- p8,l29: maybe specify: "Air temperature become as cold ..."

- p9,l14-15: please rewrite sentence

- p10,l2: wrong figure reference

- p11,l10: "there is a complicated one-to-many relationship ..." Actually it seems to be very simple: below $2\times10^{-5}$ there seems to be almost no effect, so the prewetting should just be below this value...

- p11,l14-15: I understand what is meant here, but it may be unclear for readers without a strong snow modelling background. I would explain the remeshing procedure in the model description.

- p12,l10: typo: "witch"

- p12,l21: "simulated data sets simulation" I suggest "synthetic data sets simulations"

- p13,l17: "and but does not have" please reformulate

- Appendix A: This time stepping is method is very similar to the one I used, and I based it on the work by Paniconi and Putti (1994). Maybe give them credits by citing their work?

- p14,l19: "on visual grain size measurements"

- p15,l7: "where, " —> ", where"

- p15,l11: "grain" —> "grains"

- p16,19: "is used to deal"

- p16,l24: "criteria are met"

- p16,l28: "within"

- p17: eq. A1 is not numbered as such

- p17,l13: "lower density snow that found": please reformulate.

- p17,l17: "density not included"

- p17,l19-20: please reformulate. This sentence cannot start with "while".

- Appendix B.3: Note that it should read "Daanen" and not "Dannen".

- References: a few still point to discussion papers, where final papers have already been accepted and published, for example: Avanzi et al. (2015) and Wever et al. (2016). Please provide DOIs consistently when available.

- Fig. 5: Specify here also from which date the snow profile is.

- Fig. 7: subfigure B is wrongly labelled C

**4  References**

Avanzi, F., H. Hirashima, S. Yamaguchi, T. Katsushima, and C. De Michele (2015), Laboratory-based observations of capillary barriers and preferential flow in layered snow, *Cryosphere*, *10*(5), 2013–2026, 10.5194/tc-10-2013-2016.

Calonne, N., C. Geindreau, F. Flin, S. Morin, B. Lesaffre, S. Rolland du Roscoat, and P. Charrier (2012), 3-D image-based numerical computations of snow permeability: links to specific surface area, density, and microstructural anisotropy, *Cryosphere*, *6*(5), 939–951, 10.5194/tc-6-939-2012.

DiCarlo, D. A. (2013), Stability of gravity-driven multiphase flow in porous media: 40 years of advancements, *Water Resour. Res.*, *49*(8), 4531–4544, 10.1002/wrcr.20359.

Fierz, C., R. Armstrong, Y. Durand, P. Etchevers, E. Greene, D. McClung, K. Nishimura, P. Satyawali, and S. Sokratov (2009), The International Classification for Seasonal Snow on the Ground (ICSSG), *Tech. rep.*, IHP-VII Technical Documents in Hydrology No. 83, IACS Contribution No. 1, UNESCO-IHP, Paris.

Paniconi, C. and Putti, M.: A comparison of Picard and Newton iteration in the numerical solution of multidimensional variably saturated flow problems, Water Resour. Res., 30, 3357–3374, 10.1029/94WR02046, 1994.

Techel, F. and Pielmeier, C.: Point observations of liquid water content in wet snow – investigating methodical, spatial and temporal aspects, The Cryosphere, 5, 405–418, 10.5194/tc-5-405-2011, 2011.

Vera Valero, C., N. Wever, Y. Bühler, L. Stoffel, S. Margreth, and P. Bartelt (2016), Modelling wet snow avalanche runout to assess road safety at a high-altitude mine in the central Andes, *Nat. Hazards Earth Syst. Sci.*, *16*(11), 2303–2323, 10.5194/nhess-16-2303-2016.

Wever, N., L. Schmid, A. Heilig, O. Eisen, C. Fierz, and M. Lehning (2015), Verification of the multi-layer SNOWPACK model with different water transport schemes, *Cryosphere*, *9*(6), 2271–2293, 10.5194/tc-9-2271-2015.

Wever, N., C. Vera Valero, and C. Fierz (2016), Assessing wet snow avalanche activity using detailed physics based snowpack simulations, *Geophys. Res. Lett.*, *43*(11), 5732–5740, 10.1002/2016GL068428.

Wever, N., S. Würzer, C. Fierz amd M. Lehning (2016), Simulating ice layer formation under the presence of preferential flow in layered snowpacks, *Cryosphere*, *10*(6), 2731–2744, 10.5194/tc-10-2731-2016.

Würzer, S., N. Wever, R. Juras, M. Lehning, and T. Jonas (2017), Modeling liquid water transport in snow under rain-on-snow conditions – considering preferential flow, *Hydrol. Earth Syst. Sci.*, 10.5194/hess-21-1741-2017.

———————————————————

[Figure]

**Fig. 1.** Time step histogram in Richards equation solver in the SNOWPACK model for 15 years of WFJ simulations.

[Figure]

---

## Referee Comment (RC2) · Christopher J. L. D'Amboise et al. · 30 May 2017

Overall, this is an interesting modeling paper that describes the improvements in the
Crocus(V7) snowpack model. The current modification improves the modeling of the
water storage in the snowpack, and once further validated, could be an invaluable
contribution to the state of art in snowpack modeling. Below I have a couple questions
and minor suggestions, most of which are editorial.

—

Abstract L15. Pendular and funicular regime (scientific jargon). Need to explain that
first.

Section 2. First paragraph. I am confused. Is it a three models coupled system (SUR-

FEX, ISBA, and Crocus)? If yes, then title should be changed.

The current formulation of the Richards equation 1 does not account for presence of ice and air in snowpack. How do authors think the results would change by introducing them in the equation 1?

P4. L30. h should it be H?

P5. L2,5. The notation is confusing pressure head (h), and retention curve h(theta)?

P5. L10. Equation 3. If it is water retention curve function then should be h(theta)

When water percolates through the snow layer and freezes at the bottom. The pressure and volume at the bottom grid cell increases due to ice formation. How does the model handle the increase in pressure due to ice formation?

It would be interesting to plot pressure head changes with time on Figures 8 and 9.

Section 6.6. Authors are referring to the different routines, like 'C13' and so on. It is confusing to read those notations and have no idea where they come from. For clarity, I suggest to make a chart including all the important routines.

There are figures, like Figure 11, which have the same legend. I suggest to make one legend, and put A) and B) as a subtitle or place the text inside the figure.

---

## Referee Comment (RC3) · N. Wever (Referee) · 7 Jun 2017

In the review, I used the wrong definition of the free-drainage boundary condition, using the term constant flux instead of constant hydraulic gradient. So correctly: in SNOW-PACK, we implemented free-drainage by setting the *hydraulic gradient* at the bottom of the lowest snow or soil element equal to the *hydraulic gradient* at the top of the element, while only allowing downward flux (otherwise setting flux to 0).

---

## Author Comment (AC1) · 28 Jun 2017

**Response to review:**

We would like to thank the reviewer for the constructive comments. We hope that the plots included in this document convince the reviewer that the Richards routine is acting in a numerically stable manner and the non-physical results are caused by the feedbacks discussed in the text. Therefore, we still feel that validation on lysimeter and snow pits data will not add to this work. The feedback issues are similar to the issues that the reviewer faced when developing a similar routine in the SNOWPACK model. However SNOWPACK and Crocus have different structures and slightly different equations (physical or empirical), so the issues presented are similar but unique to Crocus.

The abstract and conclusion have had major changes to highlight that the routine is complete but does not couple well in the Crocus or SURFEX model in the current state. The model code has been revised to ensure a free-flowing bottom boundary and a head based pre-wetting, as recommended by the reviewer. Relevant sections have been revised describing the new pre-wetting mechanism and the free-flowing bottom boundary. Figures 2, 3 and 6-10 have also been reproduced based on the revised model. Similarly, the results section has been revised to reflect the new bottom boundary and pre-wetting.

**Response to comments:**

For this section the authors' responses are shown in black, where the reviewer's comments are in blue. Quotes ("") and *italic font* show changes to the manuscript.

**1 general comments:**

The manuscript by d'Amboise et al. discusses the implementation of a solver for Richards equation into the detailed, multi-layer Crocus snow model. This equation describes water flow in porous media, such as snow, and previous studies have already shown that snowpack models can benefit from implementing this equation. It generally seems to provide a better representation of liquid water content (LWC) distributions and snowpack runoff behaviour. In that sense, the study represents an important step for the Crocus snow model. Also the study can be considered an important independent verification of the results achieved with the SNOWPACK model, where the solver for Richards equation has been found to considerably improve the description of liquid water flow in snow in several aspects. I found the manuscript well written and pleasant to read, but there are also some language and grammar issues (see technical corrections below). I value and appreciate the effort undertaken by the Crocus team. Based on my experience, I know that introducing new routines to a snowpack model with such a big impact as a new water percolation routine is a serious and difficult piece of work. My first main concern with the study presented by the authors is that the results presented here are not convincingly showing that the model behaves numerically stable. Distributions of liquid water content look different from distributions achieved with the SNOWPACK model. The absence of a comparison with field data of profiles of liquid water content or snowpack runoff makes it impossible to judge the validity of these results. I will do a few suggestions for additional verification of the numerical scheme, which I hope will provide convincing evidence that the model behaves numerically stable. My second main concern is that the general message of the manuscript is not clear and very open and may potentially confuse readers (see below).

We based our work off the reviewer's description of the routine that was implemented in the SNOWPACK model and described in Wever et al 2014. The review states "the study can be used as an independent verification of the results achieved with the SNOWPACK model". However there are many

small differences between the Crocus and SNOWPACK models, specifically in the model structure and some of the physics or empirical relation.

It is important to understand that issues that deal with coupling and feedback are unique to the Crocus model, but probably quite similar to issues faced when implemented in the SNOWPACK model. However, issues with parameter sets that are not a result of feedback will be common for both SNOWPACK and Crocus (grain size of crusts, strong dependence of density and grain size). Therefore, conclusions we draw from the Richards routine inside the Crocus model, may not directly apply to the Richards routine inside the SNOWPACK model.

To address the numerical stability of the routine, we turned off the two largest feedback contributions, the snow metamorphism and compaction routines (SNOWCROMETAMO & SNOWCROCOMPACTN), see Fig. A.

[Figure]

*Figure A shows results from the synthetic data set run without the compaction and metamorphism routines. Note that the snow layers are comprised of new snow grains at low density.*

Figure A shows the Richards routine behavior is stable, without any "oscillations" or the wet, dry pattern which is seen in Fig 9 & 11 of the original manuscript.

Figure B shows the same data set with the metamorphism routine still turned off, but with the compaction routine turned on.

[Figure]

*Figure B shows results from the synthetic data set run with the compaction and without the metamorphism routine.*

The oscillating pattern appears again. We show here that the pattern of wet, dry layers is not a result of a numerically unstable solution or oscillating solution in the Richards routine, but most likely an effect of an incomplete description of wet snow compaction.

Results (not shown) which used the metamorphism routine and not the compaction routine are more difficult to associate with the striped pattern. Water percolated down only after the LWC became very high > 20%. Water percolated very slow and did not reached the bottom of the snow pack by the end of the synthetic simulation, when run without compaction routine. It is very likely that the high LWCs made by the metamorphism routine will amplify the striped pattern made by the compaction routine.

The second main concern is that the message of the manuscript is not very clear. We agree that the manuscript should be more forward with the message.

We have addressed this issue by adding a few sentences to the abstract and conclusion about steps needed for further development of Crocus and the Richards routine. Together with addressing many of the points made below we feel it is clearer now.

**2 Specific comments**

I have the following remarks related to the numerical scheme:

**2.1**

Especially the alternating wet and dry snow layers shown in Fig. 9 and 11, and discussed in p10,l27, are very suspicious. It looks like a numerically oscillating solution. If it is a true LWC distribution, it is recommended to have a higher vertical resolution in the simulations (i.e., more snow layers) in order to better represent the strong gradients between the wet and dry snow layers. But the simulated values of 10%-15% seem unrealistic. Such high values may occur occasionally above capillary barriers, as shown in the work by Avanzi et al. (2016), but I'm not convinced that it should happen so regularly in the simulations shown in Fig. 9 and 11. Particularly because the artificial large snow falls create a very homogeneous stratification, such that ponding is not expected to occur. So actually I wonder if this is not a representation of the fact that the Crank-Nicolson scheme can be prone to spurious oscillations? As far as I know, Crank-Nicolson schemes are generally considered globally stable, but irregular initial conditions may lead to oscillatory behaviour. The current simulations are done with only the optimal time step for Richards equation. But are the model results sensitive to the time step inside the Richards solver? If the model is forced to run with much smaller time steps, are the results different? Or if the model is run with higher grid resolution, or by switching of remeshing, are the results different? As far as I know, the oscillations from the Crank-Nicolson scheme can be reduced by smaller time steps and/or higher grid resolution. Also stability criteria for Crank Nicolson schemes exists, which could be discussed by the authors. Note that I also have a suggestion to initialize the model more stable, see point 6 below, which may also improve numerical stability. Maybe if possible also provide additional motivation for the choice of a semi-implicit scheme instead of a fully implicit one. Is the discretization (Crank-Nicolson + Picard iteration for Richards equation) used here newly developed in this work or has it been applied before? If so, please add the references.

See section 1 (General comments), for remarks on the numerical stability and the root of the "oscillations".

A LWC of 10%- 15% is high but not unrealistic for snow (at least for short periods). However, these are the results obtained with the current state of the Crocus model using the Richards routine. We do not claim that the results of this routine accurately model snow properties, at this point. Yet, the routine has

been added to the Crocus model and further developments of the Richards routine is dependent on development of other Crocus routines and/or the parameter sets. The parameter sets have been shown to work with the SNOWPACK model. However we are cautious with them because they make up half of the feedback loop.

Normally the time step is restricted by the Crocus time step and the time step rules given in appendix A1 of the (original and revised) manuscript. To investigate the possible numerical nature of the oscillations, we have restricted the Richards time step permitting maximum values of 60 s, 10 s and 1s. This is based on the notion that numerical stability increases with decreasing time steps, but since the model operates with adjustable time steps, defining the maximum time step mainly affects those situations when the solution procedure behaves well.

Our choice for using Crank-Nicolson with the Picard iteration to solve the Richards equation comes from a previous model developed by L. Oxarango (co-author), based on the h (head) form of the Richards equation (Tinet et al. 2011). There exists an unpublished adaption which solves the mixed form of the Richards equation. The Richards equation has been solved using the Crank-Nicolson and a Picard iteration for a finite element scheme in Paniconi & Putti 1994.

**2.2**

The mass balance is verified in the Picard scheme with a threshold of $10^{-4}$. The authors should add units here, but for now I assume it is the mass balance error in m3 /m3 or kg/m2. I think that this value is set too large to judge mass conserving behaviour of the model. The minimum time step in the solver is $10^{-10}$ s (p16,l28). If the solver has a mass balance error of $10^{-4}$ with a time step of $10^{-10}$ s, this implies a mass balance error of 16 m3 /m−3 /s or 110 kg/m2 /s, or 16 kg/m2 /s, depending on the units. But this is a potentially large mass balance error! This means that if bugs in the numerical scheme or in the implementation of boundary conditions exist, the solver can "cheat" upon the mass balance check by choosing small time steps. Note that in the current version of SNOWPACK, this is also possible. During development of the solver we were particularly paying attention to the smallest time step in combination with the mass balance check. However, we relaxed the condition, by setting a low minimum time step, also allowing the solver to "cheat" the mass balance check. The motivation is to have a more robust solver for end-users. For this review, I analysed the time step distribution for running 15 years of Weissfluhjoch simulations using Richards equation with SNOWPACK, and the smallest time step during this period is about $2 \times 10^{-5}$ s. The maximum allowed mass balance error in SNOWPACK is $1 \times 10^{-10}$ kg/m2 for the entire model domain. The combination of smallest time step found in this simulation setup, together with the mass balance criterion gives $5 \times 10^{-6}$ kg/m2 /s. As can be seen in the Fig. 1 below, these small time steps happen very seldom for 15 years of simulations.

We are calculating mass balance in m because the Crocus variable holding information about wetness (PSNOWLIQ) is in m of water. Units have been added to the manuscript. The conversion to kg/m$^2$ is achieved by multiplication with the density of water (SI units).

Three mass balances are checked.

1. Full mass balance (mb_15), full snow pack over crocus (15 min) time step.
2. Full mass balance (mb_rch), full snow pack over RCH variable time step.
3. Mesh balance (mesh_bal), layer mass balance over RCH variable time step.

The convergence criteria have been evaluated for mb_rch and mesh_bal.

Figure C (mb_15) and D (time step size). We tested the performance of the models mass balance mb_15 using two limits on the mesh balance, 1E-4 m and 1E-6 m. mb_15 is important to check because it checks the results that are exported to SURFEX/ISBA and other Crocus routines.

Note: The plots have been made with the new implementation of a free flowing bottom boundary and pre-wetting based on head (h) described in sections 2.5 and 2.7 of this response.

[Figure]

*Figure C Difference in the mass balance over 15 minute Crocus time step with mesh balance limits on the Picard iteration. Red =1e-4, blue = 1e-6, pink is both red and blue.*

[Figure]

*Figure D Difference in the time step duration with mesh balance limits on the Picard iteration. Red =1e-4, blue = 1e-6, pink is both red and blue.*

The plots (C and D) show how making a stricter mass balance limit on the Picard iteration loop, has very little effect on the on the mass balance over the 15 minute crocus time step, but has a large effect on the time step size (and therefore model run time). This is because the model "cheats" the mass balance convergence criteria by reducing the time step as the reviewer points out.

The mass balance is one of the many improvements needed for this routine to be used operationally. However, we feel the mass balance is in the appropriate range when compared to the performance of the SNOWPACK model for 3 reasons (see below).

SNOWPACK has a maximum mass balance error rate of $5\times10^{-6}$ kg/m$^2$s. The rate in Crocus can be estimated from Figure C, and is $\sim$ 1E-3 kg/m$^2$ s, calculation shown below. There is a difference of 3-order of magnitude between the Crocus and SNOWPACK mass balances for their respective Richards routine.

$$m\,b_{15min} \quad \sim \frac{0.001[m] * 1000\left[\frac{kg}{m^3}\right]}{900[s]} = 0.00111\left[\frac{kg}{m^2\,s}\right]$$

In the following, we argue why this 3-order-of-magnitude difference is less problematic than it appears at a first look.

1. The Reviewer implies in (section 2.1) that in their simulations SNOWPACK does not often reach LWC in the range of 10-15%. This low water content will cause a low mass balance error which depends on the amount of water involved. In contrast, in our simulations, snow layers are often in this very wet state, hence, the associated mass balance error will be larger. This effect puts the relative difference between SNOWPACK and Crocus into perspective.

2. Our synthetic forcing has been especially designed to challenge the newly implemented Richards routine. There are very strong melt events (during day hours) after the second snow event, yet during night the upper few layers completely freeze. The routine needs to pre-wet these layers and the pressure gradient will be opposing gravity at the transition between wet and dry layers. A natural snow pack will have these conditions a few times over a season, but they are not as common or strong of a melt freeze cycle as seen in the synthetic forcing. We expect that such a demanding situation will be accompanied by a larger mass balance error compared to simulations of more naturally occurring situations

3. Crocus' melt routine sets up strong (and presumably spurious) pressure differences when the Richards routine is entered. The Crocus routine is stepped using constant time steps, during each of which, the Richards routine is entered which applies its own internal time steps. The high pressure gradients are presumably a result of prescribing a water supply mass during the first Richards time step, instead of a constant water supply rate over the entire Crocus time step. This glimpse will be fixed in future work, but is out of the scope for the current study. This would require changes to the melt routines (SNOWCROGONE, SNOWCROLAYERGONE and SNOWCROMELT), most of which would be adapting re-meshing when top layers are melted. Nevertheless, we acknowledge that this shortcoming additionally contributes to the higher mass balance error of our model.

Using a convergence criteria on mesh balance < 1E-6 m or 0.1 kg/m$^2$. Which sounds like a very relaxed criteria. However, this results in a mb_15 of $\sim$ 0.001 m (during the worst of times) often mb_15 is < 1E-6 m. Despite the three reasons given above there is only a difference of two orders of magnitude on the rate of mass balance error (kg/m$^2$ s) when compared with the SNOWPACK model.

**2.3**

A check of the second norm of the deficit vector could help to verify the correct implementation of the matrix inversion to solve the equation. Given: A . b = x, where x is the new solution of the pressure head, then the deficit vector is defined as d = A . b - x, where d is the deficit vector. It should hold that the

second norm of the vector, in case of correct implementation, is (close to) 0. Note that in an attempt to optimize the execution time of the SNOWPACK model, we removed the deficit norm check from the code, after using it first to successfully verify a correct implementation of the solver using this check. We solve A.x = b using a Thomas algorithm function. The vector d = A.x-b has been calculated with values within the range of 1E-21m to 1E-25 m. The ratio d/b (which normalizes the deficit vector) can be used as an error estimate. We have a range from ~ 1E-15 to 1E-18 (dimensionless) for d/b.

**2.4**

An overall report for the Crocus model as a whole of the mass balance may also be necessary to verify a correct implementation: Delta SWE = evaporation/condensation + sublimation/deposition + snow/rain fall + runoff?

We show above in section 2.2 of this response, that the mass balance for the Richards routine is sub-par. Therefore we do not feel the mass balance of the full Crocus model is useful at this moment.

**2.5**

With the numerical scheme for Richards equation in the SNOWPACK model, we found that an improved stability was achieved by initializing dry snow layers (the authors call it "prewetting") based on pressure head instead of LWC. The chosen form of Richards equation uses gradients in pressure head, thus it may be better to reduce gradients in pressure head when initializing the dry snow layers. Therefore, we used the procedure to determine for the whole domain the lowest pressure head in a layer corresponding to a prescribed minimum LWC. This value of the pressure head was then used to initialize dry snow layers, such that only gradients due to gravity are present. It ensures that no snow layer is initialized with a LWC above the prescribed minimum value, while at the same time starting the simulation with a numerically stable pressure head distribution.

This was a very clever hint that is now added to our routine. However the results are not noticeably different in the current state of the routine.

Text added to section 3.1 describing the new pre-wetting method (in the manuscript).

**2.6**

I noticed in the source code that C (dtheta/dh) is limited to -1−15. For what reason? C is supposed to be the exact derivative of the water retention curve. An artificial cut-off seems unnecessary and may introduce mass balance errors?

This limit has been removed. This limit was never reached during model runs that were used for the manuscript. The limit was put in place to help understanding the evolution of variables and for debugging when we had convergence troubles.

**2.7**

Section 6.2 and p16,l5-6: why not implement a free-drainage boundary condition at the bottom of the snowpack, instead of all the trouble it seems to give to use the SURFEX upper soil layer? We recently modified the SNOWPACK model such that it can run Richards equation without soil layers, and using a free drainage boundary condition seemed to work well. In SNOWPACK, we implemented freedrainage by setting the flux at the bottom of the lowest snow element similar to the flux at the top of the element, while only allowing downward flux (otherwise setting flux to 0). In case of only one snow element, we set the flux equal to the hydraulic conductivity in this element.

This is a good idea, and we have since added this to the routine with much success. We have previously unsuccessfully tried different bottom boundary conditions excluding interaction with the soil routine.

We added section 3.4.2 to the manuscript dedicated to the free flowing bottom boundary.

**3 Manuscript comments**

**3.1**

The message of the manuscript is ultimately very unclear and open. The authors apparently don't trust the new water percolation scheme enough to use it for validation (p16,l9). That basically indicates to readers that this publication is not intended to encourage users of the Crocus model to use the new percolation scheme. But what is then the main message of the manuscript? What are the next steps to improve the trust in the validity of the new routine? Are there any further developments needed or planned? The authors should also provide clear instructions of how to repeat the experiments. I was able to download the source code, but did not find any manual or readme to compile nor did it seem to contain the necessary files to run the test cases.
We have added this link to section 8 "Code and data availability" that is a walk through for running Surfex.Crocus.

https://opensource.umr-cnrm.fr/projects/snowtools/wiki/Basic_functioning_of_SURFEX_without_the_s2m_tool_from_snowtools

We do not push for the routine to be used, but rather point out where further development is needed for this routine to work well inside the SURFEX/Crocus framework. The major developments needed for this routine to be viable are outside the Richards routine, inside other Crocus routines and/or in better microstructure understanding/parameter sets. Therefore we feel that this is a complete work and should be reported on. This manuscript has two main messages:

1.  The description of the new routine.

2.  Highlighting parts of routines in Crocus that need adaptions/further development in order for the Richards routine to perform well in Crocus.

We note that the second point needs to be stated more clearly in the manuscript, and acted accordingly. Both the abstract and conclusion have significant changes.

**3.2**

I don't agree with the last sentence of the abstract. First, the absence of validation limits the value of any comment about applicability, but basically the same uncertainties in water retention curves for different snow types and for high density crusts, and also many of the feedback mechanisms are present in the SNOWPACK model too. Nevertheless, we have now demonstrated several times that solving Richards Equation is having usefull applications for the SNOWPACK model, in spite of all the uncertainties. For example for assessing wet snow stability (Wever et al., 2016a; Vera Valero et al., 2016) as well as in a detailed analysis of rain-on-snow events (Würzer et al., 2017) and for reproducing ice layers (Wever et al., 2016b). As shown in Table 1 in Wever et al. (2015), different water retention curves or different methods to determine the hydraulic conductivity at the interface nodes (arithmetic vs geometric) have limited influence on the statistics for runoff, whereas the statistics clearly improve over the bucket scheme. Therefore, I don't agree with the statement of "limited applicability" with the reasons provided in the rest of the sentence.

We can agree that Crocus and SNOWPACK are similar models that have some minor differences, however I think the minor differences can have major influence on the feedback systems in place. Both models "wet snow processes" were developed to be used with the bucket model. The calculations of "bucket size" differs between the Crocus (Vionnet et al 2012) and SNOWPACK (Wever et al. 2014) models. The bucket model for Crocus has a limit defined as a % of pore spaces (set to 5% as default, but may be changed if desired), holding capacity has a linear relationship to density. The SNOWPACK bucket size is a piecewise linear function of pore space. Essentially, Crocus has an almost binary snow wetness model with the bucket model, the SNOWPACKs bucket model seems more dynamic (in relation to pore space). This provokes the idea that wet snow processes are handled differently in SNOWPACK and Crocus.

The feedback is suspected primarily to be between the compaction & metamorphism routines and the parameterizations of the water retention curve and conductivity function. The parameterizations seem to work well inside the SNOWPACK model, despite the theoretical weaknesses that exist with them. However, we were not able to distinguish (for Crocus) if the errors come from the parameterizations (water retention curve, conductivity), other Crocus routines (compaction, metamorphism), or how they interact with each other. Since there are some theoretical problems with the parameterizations and they are the parts of the Richards routine which are part of the feedback mechanism, we cannot exclude the possibility that the parameterization used is the dominant source of error.

To address the issue, a study on microstructure is needed, either expanding the data set that the parameter sets are based on, or investigating the rates of wet snow metamorphism and compaction. Both of these experiments are outside the scope of this study.

We do not feel that our model can reproduce the results that were demonstrated in several recent publications using the SNOWPACK model applying the Richards equation. Therefore we feel the Richards routine as implemented in the SURFEX/Crocus framework has limited applicability in its current state.

The sentence in question is: "We show that the new routine has been implemented in the Crocus model, but due to amplification of parameter uncertainties through a number of feedbacks, meaningful applicability is limited until new and better parameterizations of water retention is developed for different snow types."

We do feel that this sentence should not single out the water retention curve and was changed, see below.

"*We show that the new routine has been implemented in the Crocus model, but due to feedback amplification and parameter uncertainties meaningful applicability is limited*."

**3.3**

The discussion section is nicely written and provides an interesting introspective discussion about the uncertainties and potential feedback mechanisms in water percolation modelling. Note that, however, many of the feedbacks are hypothesized or based on results of other studies. The manuscript itself does not present material supporting or quantifying the strength of those feedbacks (no validation or sensitivity study). It would be good if the discussion could be made stronger. For example, the authors may want to discuss how water retention curves for crusts potentially look like, and how strong this influences LWC distributions or snowpack runoff?

The Crocus model is not in a state where it works well with the Richards routine. However, we discuss this somewhat unexpected behavior and pinpoint the major sources of uncertainty. We demonstrate that due to the discussed feedbacks, even small structural differences (as for instance between Crocus and SNOWPACK) can be amplified leading to very different results. Furthermore, we outline potential future

improvements to avoid these problems. Altogether, we think this is a contribution worthy being reported to enable future scientific progress. However conclusions can still be drawn and supported from the work presented. Much of the discussion pertains to reinterpreting (other researcher's) experimental studies in the context of modeling, which was not done in many of the original experimental studies.

See section 3.10 (of this document) for discussion about crusts layers.

**3.4**

p4, section 2.1: Maybe explicitly state that the working of the bucket scheme in Crocus is very similar to the SNOWPACK model. p4,l9: Is the bucket size always fixed to 5% of pore space? The sentence following this sentence is a bit confusing, as if there are additional constraints. Should the sentence "For Crocus, ..." not better read "This makes the holding capacity proportional to the density of the snow layer, *but* independent of snow grain type or surrounding environment." It it also not clear what is meant by "surrounding environment" and how it could potentially influence the holding capacity?

There are no additional constraints on the "bucket size". This is stated because it is important that the reader understands that the bucket routine does not account for grain type, or the surrounding environment (layers above or below layer in question, soils water condition etc.). The Richards routine accounts for both of these (unless free flowing boundary at bottom of snow pack is used).

We have added that 5% is a default value and gave example of surrounding environment.

*"The "bucket size" or holding capacity has been defined as 5% of a layers pore spaces as default, however this can be adapted if needed. (Vionnet et al., 2012). For Crocus, the holding capacity is proportional to the density of the snow layer but is independent of snow grain type or surrounding environment (adjacent snow layers, soil, ect)."*

**3.5**

p8,l14: please provide a bit more detail on how the amount of evaporation is determined. Atmospheric forcing only provides the latent heat flux, which needs to be partitioned in evaporation and sublimation. How is Crocus doing it? Note that a reason for numerical problems with Richards equation can be when the prescribe evaporation flux exceeds the available water. For this reason SNOWPACK employs a system where the evaporation cannot exceed the amount of water available in the upper element plus the amount of water that can be advected from below given the hydraulic conductivity there. Similar for influx, although typically unrealistic large rainfall rates (>> 200 mm/hr) are necessary to exceed the absorption limit in snow of liquid water. In reality melt ponds form in snow only when liquid water cannot leave the snowpack below, not because the water input rate exceeds the snow absorption capacity.

The evaporation is described in Vionnet et al 2012 section 3.7 "Surface fluxes and surface energy balance". The evaporation is calculated before the percolation routine with the use of two routines SNOWCROBUD and SNOWCROFLUX. Vaporization/condensation can occur when the top snow layer is wet, otherwise mass flux is in a solid state (deposition/sublimation). We did not find problems with fluxes being too high. Evaporation should not affect the Richards routine in the current set up of the model, and rain rate never reaches a value close to 200 mm/ hr.

**3.6**

p13,l20: "such that the criteria to enter the percolation routine has been met." This is very confusing at it is for the first time mentioned that there are criteria whether or not to enter the percolation routine. Which criteria are meant here?

We added a sentence describing the criteria in Sec. 3.3 in the revised manuscript (Implementation of Richards routine in Crocus).

There are 2 criteria,

1. Number snow layers >= 3, currently bucket routine is used for melt out. (we do not reach melt out in this study)

2. The snowpack has to contain > pre-wetting amount of water, or have a rain rate of > $10^{-10}$

*"To reduce unnecessary computations the following 2 conditions need to be satisfied before entering the Richards routine:*
  1. *There must be a snowpack. If there is < 3 layers of snow the bucket model will be used for water percolation.*

  2. *There must be liquid water. If $\theta < \theta_{min}$, and the rain flux (over the time step of 15 min) $< 10^{-10}$ m.*

*If one of these two conditions are not satisfied Crocus will be run without calling the Richards routine"*

**3.7**

p14,l13: Many examples can be found to show that preferential flow has a much smaller typical spatial scale. See for example Fig. 2 in Techel and Pielmeier (2011), or Fig. 1a in Würzer et al. (2017). Many other examples can be found in literature.
The purpose of this sentence was to show that there is spatial variability that will affect water flow on a scale >1m. Therefore we have taken out the reference for preferential flow and adapted the sentence.

*"Hydraulic conductivity has been determined by Calonne et al., (2012) on small snow samples therefore it is possible that the measurements do not represent conductivity at a higher spatial scale because processes that affect the density distribution, grain metamorphism and pore space differ on a meter to several meter scale (Birkeland et al., 1995)."*

**3.8**

p15,l2-3: "this claim needs validation": I think it depends on the application. During the first wetting, grain shape will probably play a very important role. New snow getting wet probably retains much more liquid water initially than the water retention curve developed for melt forms will provide. For wet snow avalanche prediction, the first-wetting is often considered of crucial importance and I think improvements in the description of water flow in new snow and faceted snow (generally less shear strength) are required. On the other hand, for many hydrological applications, often the runoff behaviour during a melt season is important, for which the assumption of melt forms is justified.
The sentence in question is P15 l1 (original manuscript): "This negative feedback on snow grain type could mean the melt forms are the only crystals that need a modeled water retention curve, but this claim needs validation"

This is a matter of wet snow metamorphism speeds, which should be validated experimentally (via microstructure experiment, or a model sensitivity study). We agree that studies using a time scale of "melt season" melt forms should be sufficient. However if the rate is very fast (~30 min), melt forms may be justified for applications which use a shorter time scale.

The reviewer brings up a good point here. The bucket model has been shown to provide reliable results for runoff behavior for longer periods (Brun et al 1989 (fig 9), Brun et al 1992, (fig, 6)). Motivation to add the Richards routine to Crocus come from a desire to better simulate the first wetting of the

snowpack, and the distribution of water over the snow layers. Improvements on longer scale application are welcome, but a long term goal is to develop snowpack models to a point where first wetting can be modeled. The work done for this manuscript, does not reach this point but it is a necessary first step, from switching from the empirical routines to physical ones.

**3.9**

p15,l19: This is a bit confusing wording, as principally, I would say that snow layers are initialized with the "pre-wetting" amount. But here, it is probably meant initialization when the routine is being called during a Crocus time-step
P 15 l19 "However, 83% of snow layers were initialized in the Richards routine with a LWC< 10%"

We changed it to "…*83% of the snow layers entered the Richards routine*…"

**3.10**

p15,l26-28: Although I agree with the statement, it cannot be considered a conclusion of *this* work. It has not been demonstrated that the water flux over the crust is over- or underestimated (no validation done), neither has it been shown that the simulations are sensitive to the hydraulic parameters for crusts (no sensitivity study done)
p2, l17 we state results from Jordan (1995), that shows that waters behavior around crust layers is not well understood, some crusts promote flow and others prevent it. Therefore, the water retention curve has to have both high and low suctions when compared to surrounding snow layers. Regardless of waters behavior we are certain that the grain size used in the parameterization of the retention curve and the conductivity function is wrong, since crusts don't have individual snow grains. Because theory breaks down when it comes to crust layers there is no amount of quantifiable validation that can justify that crust are modeled in a physical way with the current parameterization. If the model is not sensitive to this error is a different question that should be validated/quantified. As far as we know this concern has not been addressed in other studies.

Our justification for using this parameterizations in the Richards routine is that there are no other parameterizations for ice crusts available.

**3.11**

Fig. 5: This figure is only mentioned once in the manuscript, and is not discussed at all. Please discuss the agreement between model and observation, or remove the snow profile.
We removed the figure.

**3.12**

I think the manuscript should not only show results for LWC distributions inside the snowpack, but also snowpack runoff

[Figure]

*Figure C Top soil layers water content for the synthetic data set, red is bucket routine, blue is Richards routine. Black arrow shows timing of 2nd snow event.*

Figure C shows a plot of the top soil layers LWC, which is not snowpack runoff but a result of snowpack runoff. The synthetic data set was designed to highlight differences between the routines over a wide range of water input rates (melt in this case). Therefore, it is not expected that the bucket routine and the Richards routine have similar flux to the soil with the synthetic data set.

This plot shows that the Richards routine is able to move water from the snowpack to the soil column. The timing of the soil wetness increase agrees well with the timing of water percolation in Fig. 9A (black arrow shows time of 2nd snow event).

**3 Technical corrections / minor comments:**

• General: note that Calonne et al. (2012) determined permeability, which can be related to conductivity. But in principle they did not publish conductivity experiments.
We have corrected this in the revised manuscript.
• General: please mention somewhere the CPU time needed to run the simulations, compared to the bucket scheme.
There are many factors that affect the run time for simulations. Using our "default" settings for this manuscript:
Convergence
head error < 1e-3 m
theta error < 1e-5 unitless
mesh bal <1e-4 m
Crocus time step of 900 s
*Output saved 900 s

\* Normal Crocus default saves output every 3 hr. or 10800 s. With Richards routine the plots look like they have randomly high pixels when the plot resolution is 3 hr. Saving the output so frequently is a major contributor to the run time.

The synthetic simulation with Richards took 9m 38s where with the bucket it took 8m 46s. Adapting the pre-wetting amount, convergences criteria (mesh_bal limit), time step and saving output resolution has a major influence over the runtime. We think it is not useful and misleading to report on the runtime of Crocus in this manuscript because many of the variables will likely change with further development of the Crocus model.

• General: there is a change of tense sometimes, compare for example section 4.3 with 3.4.

We have corrected this in the revised manuscript.

• General: sometimes "Figure" and sometimes "Fig." is used to refer to figures.

We follow the rule of GMD, Fig. should be used unless at the start of a sentence where Figure is used.

• Abstract: "this routine is based on". Why the wording "based on"? I would write here: "this routine solves Richards equation".

We have changed this in the revised manuscript.

• p2,l2-3: note that simulations using Richards Equation have already been used for the assessment of wet snow avalanche activity (Wever et al., 2016a) and for determining the initial conditions for wet snow avalanche dynamics simulations (Vera Valero et al., 2016). Given that the authors discuss this topic in particular, they may consider citing these studies here.

Below sentence with citations have been added.

*"A physically based water percolation model has been used along with weather station data to improve forecasts of wet snow stability (Wever et al., 2016a), and for determining initial conditions for simulations of avalanche dynamics (Vera Valero et al., 2016)."*

• p2,l16: This would not be the way I would explain the precondition for flow fingering, but I also have not the evidence to object against it. Maybe verify with DiCarlo (2013)? I think this is a more up-to-date citation that may be cited here aswell.

The sentence in question (below) was not changed in the modified manuscript.

"Pressure gradients acting against the gravity may induce the formation of preferential flow channels, as flow channels in soil occur where capillary pressure gradient opposes the waters flow direction (Philip, 1975)."

The reference suggested DiCarlo (2013) is a detailed review of finger flow in sand and soil with a strong focus on characterizing finger flow behavior for modeling. Since the Richards routine does not account for finger flow we do not think it is necessary to described flow fingers in such detail. The sentences purpose is to state that finger flow occurs and probably has a connection with suction (were pressure gradients oppose gravity).

We believe this is similar to the criteria used by Würzer et al. (2017) for initiating finger flow in the snowpack. However, we are unsure if the term "pressure head" in Würzer et al. (2017) refers to, pressure head or suction (negative pressure head).

• p3,l3: "Greenland Ice Sheet" (capitalized)

We have corrected this in the revised manuscript.

• p3,l8: "ice crusts": in line with the International Classification (Fierz et al., 2009), this should be "layers". The mentioned study concerned more with ice layers than with ice crusts. Similar p3,l4: lenses are discontinuous ice layers. In this case, I think it should be layers rather than lenses.

We have corrected this in the revised manuscript.

• p6,l19: "kr" should be kr.

We have corrected this in the revised manuscript.

• p7,l17: Here and elsewhere: I prefer "conductivity of snow"

We have changed this in the revised manuscript.

• p7,eq 13: I assume the second equation should read $\Delta zbot$

We have corrected this in the revised manuscript.

• p7,l14: I assume the reference should point to Eq. 13 instead of 11.

We have corrected this in the revised manuscript.

• p7,l24: citation style of Celia et al. is wrong (without parenthesis)

We have corrected this in the revised manuscript.

• p8,l3: "computation step" is a vague term. I think this refers to the iteration level k+1? Maybe write: "the pressure head h at iteration level k+1.".

We have corrected this in the revised manuscript.

• p8,l29: maybe specify: "Air temperature become as cold ..."

We have corrected this in the revised manuscript.

• p9,l14-15: please rewrite sentence

We have corrected this in the revised manuscript.

• p10,l2: wrong figure reference

We have corrected this in the revised manuscript.

• p11,l10: "there is a complicated one-to-many relationship ..." Actually it seems to be very simple: below $2 \times 10^{-5}$ there seems to be almost no effect, so the prewetting should just be below this value...

This figure and sentence have been taken out of the revised manuscript.

• p11,l14-15: I understand what is meant here, but it may be unclear for readers without a strong snow modelling background. I would explain the remeshing procedure in the model description.

"The alternating dry wet pattern appears before the second snow event when $T_{Crocus} = 60$ sec. The pattern smooths out during the second snow event, probably due to rearranging of the snow layer sizes."

These two sentence have been removed because the figure no longer shows this effect due to the head based pre-wetting and the free flowing boundary condition.

• p12,l10: typo: "witch"

We have corrected this in the revised manuscript.

• p12,l21: "simulated data sets simulation" I suggest "synthetic data sets simulations"

This sentence has been removed, because results show free-flowing bottom boundary.

• p13,l17: "and but does not have" please reformulate

This whole section has been changed.

• Appendix A: This time stepping is method is very similar to the one I used, and I based it on the work by Paniconi and Putti (1994). Maybe give them credits by citing their work?

We added this citation.

• p14,l19: "on visual grain size measurements"

We have corrected this in the revised manuscript.

• p15,l7: "where, " —> ", where"

We have corrected this in the revised manuscript.

• p15,l11: "grain" —> "grains"

We have corrected this in the revised manuscript.

• p16,l9: "is used to deal"

We have corrected this in the revised manuscript.

• p16,l24: "criteria are met"

We have corrected this in the revised manuscript.

• p16,l28: "within"

We have corrected this in the revised manuscript.

• p17: eq. A1 is not numbered as such

We have corrected this in the revised manuscript.

• p17,l13: "lower density snow that found": please reformulate.

Changed to "*For use in the Richards routine this parameter set would have to be applied to lower density snow than used to create it.*"

• p17,l17: "density not included"

We have corrected this in the revised manuscript.

• p17,l19-20: please reformulate. This sentence cannot start with "while".

While has been taken out.

• Appendix B.3: Note that it should read "Daanen" and not "Dannen".

We have corrected this in the revised manuscript.

• References: a few still point to discussion papers, where final papers have already been accepted and published, for example: Avanzi et al. (2015) and Wever et al. (2016). Please provide DOIs consistently when available.

References have been updated as needed.

• Fig. 5: Specify here also from which date the snow profile is.

Figure removed. See section 3.11 of this document.

• Fig. 7: subfigure B is wrongly labelled C

We have remade most of the figures and made sure they had correct titles and labels.

Brun, E., Martin, E., Simon, V., Gendre, C. and Coleou, C.: An Energy and Mass Model of Snow Cover Suitable for Operational Avalanche Forecasting, J. Glaciol., 35(121), 333–342, doi:10.3198/1989JoG35-121-333-342, 1989.

Brun, E., P. David, M. Sudul, and G. Brunot. "A Numerical Model to Simulate Snow-Cover Stratigraphy for Operational Avalanche Forecasting." Journal of Glaciology 38, no. 128 (January 1, 1992): 13–22. doi:10.3198/1992JoG38-128-13-22.

Birkeland, K. W., K. J. Hansen, and R. L. Brown. "The Spatial Variability of Snow Resistance on Potential Avalanche Slopes." Journal of Glaciology 41, no. 137 (1995): 183–190.

Jordan, R.: Effects of Capillary Discontinuities on Water Flow Retention in Layered Snow covers, Def. Sci. J., 45(2), 79, 1995.

Paniconi, Claudio, and Mario Putti. "A Comparison of Picard and Newton Iteration in the Numerical Solution of Multidimensional Variably Saturated Flow Problems." *Water Resources Research* 30, no. 12 (1994): 3357–3374.

Tinet, A. J., L. Oxarango, R. Bayard, H. Benbelkacem, Guillaume Stoltz, M. J. Staub, and J.-P. Gourc. "Experimental and Theoretical Assessment of the Multi-Domain Flow Behaviour in a Waste Body during Leachate Infiltration." *Waste Management* 31, no. 8 (2011): 1797–1806.

Vionnet, V., Brun, E., Morin, S., Boone, A., Faroux, S., Le Moigne, P., Martin, E. and Willemet, J.-M.: The detailed snowpack scheme Crocus and its implementation in SURFEX v7.2, Geosci Model Dev, 5(3), 773–791, doi:10.5194/gmd-5-773-2012, 2012.

Würzer, Sebastian, Wever, Nander, Juras, Roman, Lehning, Roman, and Jonas, Tobias, "Modelling Liquid Water Transport in Snow under Rain-on-Snow Conditions-Considering Preferential Flow." Hydrology and Earth System Sciences 21, no. 3: 1741, 2017

Wever, N., Fierz, C., Mitterer, C., Hirashima, H. and Lehning, M.: Solving Richards Equation for snow improves snowpack meltwater runoff estimations in detailed multi-layer snowpack model, The Cryosphere, 8(1), 257–274, doi:10.5194/tc-8-257-2014, 2014.

---

## Author Comment (AC2) · 28 Jun 2017

**Response to review:**

We would like to thank the reviewer for the comments.

**Response to comments:**

For this section the authors response are shown in black, where the reviewers comments are in blue. Quotes ("") and *italic font* show changes to the manuscript.

Overall, this is an interesting modeling paper that describes the improvements in the Crocus(V7) snowpack model. The current modification improves the modeling of the water storage in the snowpack, and once further validated, could be an invaluable contribution to the state of art in snowpack modeling. Below I have a couple questions and minor suggestions, most of which are editorial.

Abstract L15. Pendular and funicular regime (scientific jargon). Need to explain that first.
We added to the abstract to further explain the jargon.

*"Snow layers often reached a point where the ice crystals surface area is completely covered by a thin film of water (the transition between pendular and funicular regimes), where feedback is expected to be nonlinear.."*

Section 2. First paragraph. I am confused. Is it a three models coupled system (SURFEX, ISBA, and Crocus)? If yes, then title should be changed.
The Crocus model has been developed as a standalone model. However, Crocus is often run coupled with two other models (SURFEX and ISBA). Crocus was coupled to "Interactions between soil, biosphere and atmosphere" (ISBA), for a wider range of bottom boundary conditions (Vionnet et al 2012, Noilhan and Planton, 1989).  ISBA is coupled to SURFEX for a similar reason regarding the upper boundary.

Because the changes are all contained in the Crocus model we feel the title should include Crocus, but have added SURFEX to the title. We have adapted the 1st paragraph on section 2 of the revised manuscript to make connections between the models more obvious.

The current formulation of the Richards equation 1 does not account for presence of ice and air in snowpack. How do authors think the results would change by introducing them in the equation 1?
The Richards equation does account for ice, air and water in the snowpack via the water retention curve and the hydraulic conductivity. Both of these parameterizations are based on the snow layers dry density (ice vs air) the snow grains size (distribution of ice and air) and $\theta$, the water content.

P4. L30. h should it be H?
Equation 2. shows the relationship between H and h.

*"H is the hydraulic head, which is the sum of the pressure head (h) and the elevation (z), which is negative because z is positive downward (Eq. 2)."*
$$H = h - z \qquad \qquad \text{Equation 2}$$

P5. L2,5. The notation is confusing pressure head (h), and retention curve h(theta)?

We took out h(θ), to avoid confusion. The water retention curve relates head values (h) to water content values (θ) and vice-versa.

Equation 3 shows θ(h) which is the inverse function of h(θ). We use the relations θ(h) and h(θ) in our routine. The Van Genuchten (1980) (VG) parameterization is used for the relation between water content and pressure head which can be seen in Fig.2 . Figure 2 shows for a given grain size and snow layer density there is a one to one relation between h and θ. We now use only "θ" and "h", to make notation clearer.

When water percolates through the snow layer and freezes at the bottom. The pressure and volume at the bottom grid cell increases due to ice formation. How does the model handle the increase in pressure due to ice formation?
This behavior described by the reviewer can be seen in the Filefjell results (Fig. 6 D of the revised manuscript). In the Filefjell case the ice layer at the bottom of the snowpack becomes very dense. The model will eventually crash due to extreme values of the suction and conductivity if the density becomes too great.

I assume "increase in pressure" means increase in suction (negative pressure), since this model does not deal with positive pressures (no pooling of water). Regardless of the terminology it is the pressure gradient (dh/dz) what matters for water percolation. It is not obvious what will happen when one layer becomes denser. The hydraulic conductivity and the water retention curve will both be affected by the change in density and also the change in θ because the pore spaces decrease.

In general it is not well understood how water interacts with crust layers it has been reported to act as a barrier layer but also a conduit (Jordan, 1995). Better parameter sets are needed for crust layers as stated in section 6.5.
We feel very dense layers will not be problematic if new parameter sets for crusts are developed and feedback from the snow compaction routine is updated for high water contents.

It would be interesting to plot pressure head changes with time on Figures 8 and 9.
The saturation (S defined in equation 6) is the major factor in a snow layers pressure head. This can be seen in Fig. 2. Snow density and grain size have a smaller effect on the pressure head of the snow pack. Because the pressure changes so drastically with regards to S (or θ), we feel it is better to show how θ changes with time and the density evolution instead of changes in h. The histogram in Fig. 2 shows how pressure head acts at different water contents, this is particularly interesting to see that the saturation is always < 20% for the simulated forcing.

Section 6.6. Authors are referring to the different routines, like 'C13' and so on. It is confusing to read those notations and have no idea where they come from. For clarity, I suggest to make a chart including all the important routines.
C13 and B92 are parts of the SNOWCROMETAMO routine shown in figure 1. We make this clearer in section 6.5 by describing the relationship between C13, B92 and the routine SNOWCROMETAMO. We also refer to figure 1.

*"The water retention curve and hydraulic conductivity functions are not designed for use on crust layers. Furthermore, Crocus' snow metamorphism routine (SNOWCROMETAMO, Fig. 1) does not work well for*

*crust layers, because dense crusts do not have individual grains, but rather a solid ice layer with bubbles. There is a choice of routines in SNOWCROMETAMO, the B92 (Brun et al. 1992) and the C13 routines. The B92 routine uses sphericity and dendricity. The C13 routine uses optical diameter, which is used as an approximation for visual grain size in the Richards routine."*

There are figures, like Figure 11, which have the same legend. I suggest to make one legend, and put A) and B) as a subtitle or place the text inside the figure

Figure 11 (Fig. 9 in revised manuscript) has been updated with A and B and the time step duration printed inside the plot area.
"

[Figure]

*Figure 9: Water percolation in the simulated data set scenario with different time steps: A) $T_{Crocus}$= 900 s (15 min), B) $T_{Crocus}$= 60 s both plots use the free-flowing bottom boundary with pre-wetting set to $\theta_{min} = 10^{-5}$. The 60s simulation has water at the bottom of the snowpack 4 days before the 900 s simulation. "*

**References**

Brun, E., David, P., Sudul, M., and Brunot. G.: A Numerical Model to Simulate Snow-Cover Stratigraphy for Operational Avalanche Forecasting. J Glaciol 38(128), 13–22. doi:10.3198/1992JoG38-128-13-22., 1992

Jordan, R.: Effects of Capillary Discontinuities on Water Flow Retention in Layered Snow covers, Def. Sci. J., 45(2), 79, 1995.

Noilhan, J. and Planton, S.: A simple parameterization of land surface processes for meteorological models, Mon. Wea. Rev., 117, 536–549, 1989

Van Genuchten, M. T.: A closed-form equation for predicting the hydraulic conductivity of unsaturated soils, Soil Sci. Soc. Am. J., 44(5), 892–898, 1980.

Vionnet, V., Brun, E., Morin, S., Boone, A., Faroux, S., Le Moigne, P., Martin, E. and Willemet, J.-M.: The detailed snowpack scheme Crocus and its implementation in SURFEX v7.2, Geosci Model Dev, 5(3), 773–791, doi:10.5194/gmd-5-773-2012, 2012.

Vionnet, V., Brun, E., Morin, S., Boone, A., Faroux, S., Le Moigne, P., Martin, E. and Willemet, J.-M.: The detailed snowpack scheme Crocus and its implementation in SURFEX v7.2, Geosci Model Dev, 5(3), 773–791, doi:10.5194/gmd-5-773-2012, 2012.

---

## Author Response (AR1)

N. Wever (Referee)

nander.wever@slf.ch

10 Received and published: 2 May 2017

1 General comments

The manuscript by d'Amboise et al. discusses the implementation of a solver for Richards equation into the detailed, multi-layer Crocus snow model. This equation describes water flow in porous media, such as snow, and previous studies have already

- 15 shown that snowpack models can benefit from implementing this equation. It generally seems to provide a better representation of liquid water content (LWC) distributions and snowpack runoff behaviour. In that sense, the study represents an important step for the Crocus snow model. Also the study can be considered an important indepenC1 dent verification of the results achieved with the SNOWPACK model, where the solver
- 20 for Richards equation has been found to considerably improve the description of liquid water flow in snow in several aspects. I found the manuscript well written and pleasant to read, but there are also some language and grammar issues (see technical corrections below). I value and appreciate the effort undertaken by the Crocus team. Based

on my experience, I know that introducing new routines to a snowpack model with such a big impact as a new water percolation routine is a serious and difficult piece of work. My first main concern with the study presented by the authors is that the results presented here are not convincingly showing that the model behaves numerically stable.

- 5 Distributions of liquid water content look different from distributions achieved with the SNOWPACK model. The absence of a comparison with field data of profiles of liquid water content or snowpack runoff makes it impossible to judge the validity of these results. I will do a few suggestions for additional verification of the numerical scheme, which I hope will provide convincing evidence that the model behaves numerically stable.
- 10 My second main concern is that the general message of the manuscript is not clear and very open and may potentially confuse readers (see below).

2 Specific comments

I have the following remarks related to the numerical scheme:

1. Especially the alternating wet and dry snow layers shown in Fig. 9 and 11, and

- 15 discussed in p10,127, are very suspicious. It looks like a numerically oscillating solution. If it is a true LWC distribution, it is recommended to have a higher vertical resolution in the simulations (i.e., more snow layers) in order to better represent the strong gradients between the wet and dry snow layers. But the simulated values of 10%-15% seem unrealistic. Such high values may occur ocC2
- 20 casionally above capillary barriers, as shown in the work by Avanzi et al. (2016), but I'm not convinced that it should happen so regularly in the simulations shown in Fig. 9 and 11. Particularly because the artificial large snow falls create a very homogeneous stratification, such that ponding is not expected to occur. So actually

I wonder if this is not a representation of the fact that the Crank-Nicolson scheme can be prone to spurious oscillations? As far as I know, Crank-Nicolson schemes are generally considered globally stable, but irregular initial conditions may lead to oscillatory behaviour. The current simulations are done with only the

- 5 optimal time step for Richards equation. But are the model results sensitive to the time step inside the Richards solver? If the model is forced to run with much smaller time steps, are the results different? Or if the model is run with higher grid resolution, or by switching of remeshing, are the results different? As far as I know, the oscillations from the Crank-Nicolson scheme can be reduced by
- 10 smaller time steps and/or higher grid resolution. Also stability criteria for Crank Nicolson schemes exists, which could be discussed by the authors. Note that I also have a suggestion to initialize the model more stable, see point 6 below, which may also improve numerical stability. Maybe if possible also provide additional motivation for the choice of a semi-implicit scheme instead of a fully implicit
- 15 one. Is the discretization (Crank-Nicolson + Picard iteration for Richards equation) used here newly developed in this work or has it been applied before? If so, please add the references.
  - 2. The mass balance is verified in the Picard scheme with a threshold of 10–4 . The
- 20 authors should add units here, but for now I assume it is the mass balance error in m3

/m3 or kg/m2

. I think that this value is set too large to judge mass conserving

behaviour of the model. The minimum time step in the solver is 10–10 s (p16,l28). If the solver has a mass balance error of 10–4 with a time step of 10–10 s, this implies a mass balance error of 16 m3

/m-3

5 /s or 110 kg/m2

/s, or 16 kg/m2

/s, depending

on the units. But this is a potentially large mass balance error! This

- 10 means that if bugs in the numerical scheme or in the implementation of boundary conditions exist, the solver can "cheat" upon the mass balance check by choosing small time steps. Note that in the current version of SNOWPACK, this is also possible. During development of the solver we were particularly paying attention to the smallest time step in combination with the mass balance check. However,
- 15 we relaxed the condition, by setting a low minimum time step, also allowing the solver to "cheat" the mass balance check. The motivation is to have a more robust solver for end-users. For this review, I analysed the time step distribution for running 15 years of Weissfluhjoch simulations using Richards equation with SNOWPACK, and the smallest time step during this period is about 2x10–5 s.
- 20 The maximum allowed mass balance error in SNOWPACK is 1x10–10 kg/m2 for

the entire model domain. The combination of smallest time step found in this simulation setup, together with the mass balance criterion gives 5x10-6 kg/m2

As can be seen in the Fig. 1 below, these small time steps happen very seldom for 15 years of simulations.

3. A check of the second norm of the deficit vector could help to verify the correct

- 5 implementation of the matrix inversion to solve the equation. Given:  $A \cdot b = x$ , where x is the new solution of the pressure head, then the deficit vector is defined as  $d = A \cdot b - x$ , where d is the deficit vector. It should hold that the second norm of the vector, in case of correct implementation, is (close to) 0. Note that in an attempt to optimize the execution time of the SNOWPACK model, we removed
- 10 the deficit norm check from the code, after using it first to successfully verify a correct implementation of the solver using this check.

4. An overall report for the Crocus model as a whole of the mass balance may
also be necessary to verify a correct implementation: Delta SWE = evaporation/condensation
+ sublimation/deposition + snow/rain fall + runoff?

15 5. With the numerical scheme for Richards equation in the SNOWPACK model, weC4

found that an improved stability was achieved by initializing dry snow layers (the authors call it "prewetting") based on pressure head instead of LWC. The chosen form of Richards equation uses gradients in pressure head, thus it may be

20 better to reduce gradients in pressure head when initializing the dry snow layers. Therefore, we used the procedure to determine for the whole domain the lowest pressure head in a layer corresponding to a prescribed minimum LWC. This value of the pressure head was then used to initialize dry snow layers, such that only gradients due to gravity are present. It ensures that no snow layer is initialized with a LWC above the prescribed minimum value, while at the same time starting the simulation with a numerically stable pressure head distribution.

6. I noticed in the source code that C (dtheta/dh) is limited to -1-15. For what reason?

- 5 C is supposed to be the exact derivative of the water retention curve. An artificial cut-off seems unnecessary and may introduce mass balance errors?
  7. Section 6.2 and p16,15-6: why not implement a free-drainage boundary condition at the bottom of the snowpack, instead of all the trouble it seems to give to use the SURFEX upper soil layer? We recently modified the SNOWPACK model such
- 10 that it can run Richards equation without soil layers, and using a free drainage boundary condition seemed to work well. In SNOWPACK, we implemented freedrainage by setting the flux at the bottom of the lowest snow element similar to the flux at the top of the element, while only allowing downward flux (otherwise setting flux to 0). In case of only one snow element, we set the flux equal to the
- 15 hydraulic conductivity in this element.

I have the following remarks regarding the manuscript itself:

1. The message of the manuscript is ultimately very unclear and open. The authors apparently don't trust the new water percolation scheme enough to use it for validation (p16,19). That basically indicates to readers that this publication is not

```
20 C5
```

intended to encourage users of the Crocus model to use the new percolation scheme. But what is then the main message of the manuscript? What are the next steps to improve the trust in the validity of the new routine? Are there any further developments needed or planned? The authors should also provide clear instructions of how to repeat the experiments. I was able to download the source code, but did not find any manual or readme to compile nor did it seem to contain the necessary files to run the test cases.

- 5 2. I don't agree with the last sentence of the abstract. First, the absence of validation limits the value of any comment about applicability, but basically the same uncertainties in water retention curves for different snow types and for high density crusts, and also many of the feedback mechanisms are present in the SNOWPACK model too. Nevertheless, we have now demonstrated several times
- 10 that solving Richards Equation is having usefull applications for the SNOWPACK model, in spite of all the uncertainties. For example for assessing wet snow stability (Wever et al., 2016a; Vera Valero et al., 2016) as well as in a detailed analysis of rain-on-snow events (Würzer et al., 2017) and for reproducing ice layers (Wever et al., 2016b). As shown in Table 1 in Wever et al. (2015), different
- 15 water retention curves or different methods to determine the hydraulic conductivity at the interface nodes (arithmetic vs geometric) have limited influence on the statistics for runoff, whereas the statistics clearly improve over the bucket scheme. Therefore, I don't agree with the statement of "limited applicability" with the reasons provided in the rest of the sentence.
- 20 3. The discussion section is nicely written and provides an interesting introspective discussion about the uncertainties and potential feedback mechanisms in water percolation modelling. Note that, however, many of the feedbacks are hypothesized or based on results of other studies. The manuscript itself does not present

material supporting or quantifying the strength of those feedbacks (no validation or sensitivity study). It would be good if the discussion could be made stronger.

For example, the authors may want to discuss how water retention curves for

- 5 crusts potentially look like, and how strong this influences LWC distributions or snowpack runoff?
  - 4. p4, section 2.1: Maybe explicitly state that the working of the bucket scheme in Crocus is very similar to the SNOWPACK model. p4,19: Is the bucket size always fixed to 5% of pore space? The sentence following this sentence is a
- 10 bit confusing, as if there are additional constraints. Should the sentence "For Crocus, ..." not better read "This makes the holding capacity proportional to the density of the snow layer, \*but\* independent of snow grain type or surrounding environment." It it also not clear what is meant by "surrounding environment" and how it could potentially influence the holding capacity?
- 15 5. p8,114: please provide a bit more detail on how the amount of evaporation is determined. Atmospheric forcing only provides the latent heat flux, which needs to be partitioned in evaporation and sublimation. How is Crocus doing it? Note that a reason for numerical problems with Richards equation can be when the prescribe evaporation flux exceeds the available water. For this reason SNOWPACK
- 20 employs a system where the evaporation cannot exceed the amount of water available in the upper element plus the amount of water that can be advected from below given the hydraulic conductivity there. Similar for influx, although typically unrealistic large rainfall rates (>> 200 mm/hr) are necessary to exceed the

absorption limit in snow of liquid water. In reality melt ponds form in snow only when liquid water cannot leave the snowpack below, not because the water input rate exceeds the snow absorption capacity.

6. p13,l20: "such that the criteria to enter the percolation routine has been met."

5 This is very confusing at it is for the first time mentioned that there are criteria whether or not to enter the percolation routine. Which criteria are meant here?
7. p14,113: Many examples can be found to show that preferential flow has a much C7

smaller typical spatial scale. See for example Fig. 2 in Techel and Pielmeier

10 (2011), or Fig. 1a in Würzer et al. (2017). Many other examples can be found in literature.

8. p15,12-3: "this claim needs validation": I think it depends on the application. During the first wetting, grain shape will probably play a very important role. New snow getting wet probably retains much more liquid water initially than the water

- 15 retention curve developed for melt forms will provide. For wet snow avalanche prediction, the first-wetting is often considered of crucial importance and I think improvements in the description of water flow in new snow and faceted snow (generally less shear strength) are required. On the other hand, for many hydrological applications, often the runoff behaviour during a melt season is important,
- 20 for which the assumption of melt forms is justified.

9. p15,119: This is a bit confusing wording, as principally, I would say that snow layers are initialized with the "pre-wetting" amount. But here, it is probably meant initialization when the routine is being called during a Crocus time-step.

10. p15,126-28: Although I agree with the statement, it cannot be considered a conclusion of \*this\* work. It has not been demonstrated that the water flux over the crust is over- or underestimated (no validation done), neither has it been shown that the simulations are sensitive to the hydraulic parameters for crusts (no sensitivity

5 study done).

11. Fig. 5: This figure is only mentioned once in the manuscript, and is not discussed at all. Please discuss the agreement between model and observation, or remove the snow profile.

12. I think the manuscript should not only show results for LWC distributions inside

10 the snowpack, but also snowpack runoff.

**C8**

3 Technical corrections / minor comments:

• General: note that Calonne et al. (2012) determined permeability, which can be related to conductivity. But in principle they did not publish conductivity experiments.

 General: please mention somewhere the CPU time needed to run the simulations, compared to the bucket scheme.

• General: there is a change of tense sometimes, compare for example section 4.3 with 3.4.

• General: sometimes "Figure" and sometimes "Fig." is used to refer to figures.

 Abstract: "this routine is based on". Why the wording "based on"? I would write here: "this routine solves Richards equation".

• p2,l2-3: note that simulations using Richards Equation have already been used for the assessment of wet snow avalanche activity (Wever et al., 2016a) and for

determining the initial conditions for wet snow avalanche dynamics simulations (Vera Valero et al., 2016). Given that the authors discuss this topic in particular, they may consider citing these studies here.

• p2,116: This would not be the way I would explain the precondition for flow fingering,

5 but I also have not the evidence to object against it. Maybe verify with

DiCarlo (2013)? I think this is a more up-to-date citation that may be cited here aswell.

• p3,l3: "Greenland Ice Sheet" (capitalized)

- p3,18: "ice crusts": in line with the International Classification (Fierz et al., 2009),
- 10 this should be "layers". The mentioned study concerned more with ice layers than

with ice crusts. Similar p3,14: lenses are discontinuous ice layers. In this case, I think it should be layers rather than lenses.

• p6,119: "kr" should be kr.

- 15 p7,117: Here and elsewhere: I prefer "conductivity of snow"
  - p7,eq 13: I assume the second equation should read  $\Delta z$ bot
  - p7,114: I assume the reference should point to Eq. 13 instead of 11.
  - p7,124: citation style of Celia et al. is wrong (without parenthesis)
  - p8,l3: "computation step" is a vague term. I think this refers to the iteration level
- 20 k+1? Maybe write: "the pressure head h at iteration level k+1.".
  - p8,129: maybe specify: "Air temperature become as cold ..."
  - p9,114-15: please rewrite sentence
  - p10,l2: wrong figure reference

• p11,110: "there is a complicated one-to-many relationship ..." Actually it seems to be very simple: below 2x10–5 there seems to be almost no effect, so the

prewetting should just be below this value ...

p11,l14-15: I understand what is meant here, but it may be unclear for readers without a strong snow modelling background. I would explain the remeshing procedure in the model description.

• p12,110: typo: "witch"

• p12,l21: "simulated data sets simulation" I suggest "synthetic data sets simulations"

• p13,117: "and but does not have" please reformulate

• Appendix A: This time stepping is method is very similar to the one I used, and

I based it on the work by Paniconi and Putti (1994). Maybe give them credits by

citing their work?

- 15 p14,119: "on visual grain size measurements"
  - p15,l7: "where, " —> ", where"
  - p15,l11: "grain" —> "grains"
  - p16,19: "is used to deal"
  - p16,l24: "criteria are met"
- 20 p16,l28: "within"
  - p17: eq. A1 is not numbered as such
  - p17,113: "lower density snow that found": please reformulate.
  - p17,l17: "density not included"

- p17,119-20: please reformulate. This sentence cannot start with "while".
- Appendix B.3: Note that it should read "Daanen" and not "Dannen".
- References: a few still point to discussion papers, where final papers have already been accepted and published, for example: Avanzi et al. (2015) and Wever
- 5 et al. (2016). Please provide DOIs consistently when available.
  - Fig. 5: Specify here also from which date the snow profile is.
  - Fig. 7: subfigure B is wrongly labelled C
  - C11

4 References

 Avanzi, F., H. Hirashima, S. Yamaguchi, T. Katsushima, and C. De Michele (2015), Laboratory-based observations of capillary barriers and preferential flow in layered snow, Cryosphere, 10(5), 2013–2026, 10.5194/tc-10-2013-2016.

Calonne, N., C. Geindreau, F. Flin, S. Morin, B. Lesaffre, S. Rolland du Roscoat, and P. Charrier (2012), 3-D image-based numerical computations of snow permeability:

links to specific surface area, density, and microstructural anisotropy, Cryosphere, 6(5),
939–951, 10.5194/tc-6-939-2012.

DiCarlo, D. A. (2013), Stability of gravity-driven multiphase flow in porous media: 40
years of advancements, Water Resour. Res., 49(8), 4531–4544, 10.1002/wrcr.20359.
Fierz, C., R. Armstrong, Y. Durand, P. Etchevers, E. Greene, D. McClung, K. Nishimura,

 P. Satyawali, and S. Sokratov (2009), The International Classification for Seasonal Snow on the Ground (ICSSG), Tech. rep., IHP-VII Technical Documents in Hydrology No. 83, IACS Contribution No. 1, UNESCO-IHP, Paris.

Paniconi, C. and Putti, M.: A comparison of Picard and Newton iteration in the numerical

solution of multidimensional variably saturated flow problems, Water Resour. Res.,

30, 3357–3374, 10.1029/94WR02046, 1994.

Techel, F. and Pielmeier, C.: Point observations of liquid water content in wet snow – investigating methodical, spatial and temporal aspects, The Cryosphere, 5, 405–418,

5 10.5194/tc-5-405-2011, 2011.

Vera Valero, C., N. Wever, Y. Bühler, L. Stoffel, S. Margreth, and P. Bartelt (2016), Modelling wet snow avalanche runout to assess road safety at a high-altitude mine in the central Andes, Nat. Hazards Earth Syst. Sci., 16(11), 2303–2323, 10.5194/nhess-16-2303-2016.

```
10 C12
```

Wever, N., L. Schmid, A. Heilig, O. Eisen, C. Fierz, and M. Lehning (2015), Verification of the multi-layer SNOWPACK model with different water transport schemes, Cryosphere, 9(6), 2271–2293, 10.5194/tc-9-2271-2015.

Wever, N., C. Vera Valero, and C. Fierz (2016), Assessing wet snow avalanche activity

using detailed physics based snowpack simulations, Geophys. Res. Lett., 43(11),
5732–5740, 10.1002/2016GL068428.

Wever, N., S. Würzer, C. Fierz amd M. Lehning (2016), Simulating ice layer formation under the presence of preferential flow in layered snowpacks, Cryosphere, 10(6), 2731–2744, 10.5194/tc-10-2731-2016.

Würzer, S., N. Wever, R. Juras, M. Lehning, and T. Jonas (2017), Modeling liquid water transport in snow under rain-on-snow conditions – considering preferential flow, Hydrol. Earth Syst. Sci., 10.5194/hess-21-1741-2017.

Fig. 1. Time step histogram in Richards equation solver in the SNOWPACK model for 15 years of WFJ simulations.

**5 Review 1.1**

Interactive comment on "Implementation of a

physically based water percolation routine in the

Crocus (V7) snowpack model" by

Christopher J. L. D'Amboise et al.

**10 N. Wever (Referee)**

nander.wever@slf.ch

Received and published: 7 June 2017

In the review, I used the wrong definition of the free-drainage boundary condition, using

the term constant flux instead of constant hydraulic gradient. So correctly: in SNOWPACK,

we implemented free-drainage by setting the hydraulic gradient at the bottom of the lowest snow or soil element equal to the hydraulic gradient at the top of the element, while only allowing downward flux (otherwise setting flux to 0). Interactive comment on Geosci. Model Dev. Discuss., https://doi.org/10.5194/gmd-2017-56, 2017.

20 C1

**Review 2**

Geosci. Model Dev. Discuss.,
doi:10.5194/gmd-2017-56-RC2, 2017
© Author(s) 2017. CC-BY 3.0 License.
5 Interactive comment on "Implementation of a physically based water percolation routine in the Crocus (V7) snowpack model" by
Christopher J. L. D'Amboise et al.
Anonymous Referee #2
10 Received and published: 30 May 2017
Overall, this is an interesting modeling paper that describes the improvements in the

Crocus(V7) snowpack model. The current modification improves the modeling of the water storage in the snowpack, and once further validated, could be an invaluable contribution to the state of art in snowpack modeling. Below I have a couple questions

15 and minor suggestions, most of which are editorial.

Abstract L15. Pendular and funicular regime (scientific jargon). Need to explain that first.

Section 2. First paragraph. I am confused. Is it a three models coupled system (SURC1

20 FEX, ISBA, and Crocus)? If yes, then title should be changed.

The current formulation of the Richards equation 1 does not account for presence of ice and air in snowpack. How do authors think the results would change by introducing them in the equation 1?

P4. L30. h should it be H?

P5. L2,5. The notation is confusing pressure head (h), and retention curve h(theta)? P5. L10. Equation 3. If it is water retention curve function then should be h(theta) When water percolates through the snow layer and freezes at the bottom. The pressure

5 and volume at the bottom grid cell increases due to ice formation. How does the model handle the increase in pressure due to ice formation?

It would be interesting to plot pressure head changes with time on Figures 8 and 9. Section 6.6. Authors are referring to the different routines, like 'C13' and so on. It is confusing to read those notations and have no idea where they come from. For clarity,

10 I suggest to make a chart including all the important routines.

There are figures, like Figure 11, which have the same legend. I suggest to make one

legend, and put A) and B) as a subtitle or place the text inside the figure.

15

**SECTION 2:** Author's response**

**Reply to review 1**

**Response to review:**

20

We would like to thank the reviewer for the constructive comments. We hope that the plots included in this document convince the reviewer that the Richards routine is acting in a numerically stable manner and the non-physical results are caused by the feedbacks discussed in the text. Therefore, we still feel that validation on lysimeter and snow pits data will not add to this work. The feedback issues are similar to the issues that the reviewer faced when developing a similar routine in the SNOWPACK model. However SNOWPACK and Crocus have different structures and slightly different equations (physical or empirical), so the issues presented are similar but unique to Crocus.

The abstract and conclusion have had major changes to highlight that the routine is complete but does not couple well in the Crocus or SURFEX model in the current state. The model code has been revised to ensure a free-flowing bottom boundary and a head based pre-wetting, as recommended by the reviewer. Relevant sections have been revised describing the new pre-

5 wetting mechanism and the free-flowing bottom boundary. Figures 2, 3 and 6-10 have also been reproduced based on the revised model. Similarly, the results section has been revised to reflect the new bottom boundary and pre-wetting.

**Response to comments:**

For this section the authors' responses are shown in black, where the reviewer's comments are in blue. Quotes ("") and *italic font* show changes to the manuscript.

**10**

**1 general comments:**

The manuscript by d'Amboise et al. discusses the implementation of a solver for Richards equation into the detailed, multilayer Crocus snow model. This equation describes water flow in porous media, such as snow, and previous studies have already shown that snowpack models can benefit from implementing this equation. It generally seems to provide a better representation

- 15 of liquid water content (LWC) distributions and snowpack runoff behaviour. In that sense, the study represents an important step for the Crocus snow model. Also the study can be considered an important independent verification of the results achieved with the SNOWPACK model, where the solver for Richards equation has been found to considerably improve the description of liquid water flow in snow in several aspects. I found the manuscript well written and pleasant to read, but there are also some language and grammar issues (see technical corrections below). I value and appreciate the effort undertaken by the
- 20 Crocus team. Based on my experience, I know that introducing new routines to a snowpack model with such a big impact as a new water percolation routine is a serious and difficult piece of work. My first main concern with the study presented by the authors is that the results presented here are not convincingly showing that the model behaves numerically stable. Distributions of liquid water content look different from distributions achieved with the SNOWPACK model. The absence of a comparison with field data of profiles of liquid water content or snowpack runoff makes it impossible to judge the validity of these results.
- 25 I will do a few suggestions for additional verification of the numerical scheme, which I hope will provide convincing evidence that the model behaves numerically stable. My second main concern is that the general message of the manuscript is not clear and very open and may potentially confuse readers (see below).

We based our work off the reviewer's description of the routine that was implemented in the SNOWPACK model and described in Wever et al 2014. The review states "the study can be used as an independent verification of the results achieved with the SNOWPACK model". However there are many small differences between the Crocus and SNOWPACK models, specifically in the model structure and some of the physics or empirical relation.

- 5 It is important to understand that issues that deal with coupling and feedback are unique to the Crocus model, but probably quite similar to issues faced when implemented in the SNOWPACK model. However, issues with parameter sets that are not a result of feedback will be common for both SNOWPACK and Crocus (grain size of crusts, strong dependence of density and grain size). Therefore, conclusions we draw from the Richards routine inside the Crocus model, may not directly apply to the Richards routine inside the SNOWPACK model.
- 10 To address the numerical stability of the routine, we turned off the two largest feedback contributions, the snow metamorphism and compaction routines (SNOWCROMETAMO & SNOWCROCOMPACTN), see Fig. A.

Figure A shows results from the synthetic data set run without the compaction and metamorphism routines. Note that the snow layers are comprised of new snow grains at low density.

Figure A shows the Richards routine behavior is stable, without any "oscillations" or the wet, dry pattern which is seen in Fig

15 9 & 11 of the original manuscript.

Figure B shows the same data set with the metamorphism routine still turned off, but with the compaction routine turned on.

---

## Author Response (AR2)

**Suggestions for revision or reasons for rejection (will be published if the paper is accepted for final publication)**

The authors provided an extensive response-to-review document, which is really appreciated, as well as a thoroughly revised manuscript. I'm very impressed with the response to the review provided by the authors, because it provided many insights into the new percolation scheme in CROCUS. And to many issues raised by the reviewers, the responses were very informative and clear. However, I'm a little bit disappointed that some of the interesting points in this document did not end up in the revised manuscript, although I think that by doing so, the significance of the manuscript would be enhanced and it would result into a much more interesting manuscript. Please find detailed comments below. I consider point 1, 2 and 5 are of particular importance to be addressed adequately, after which I can recommend publication.

Thank you for the second in-depth review. We have revised the manuscript considering the points brought to our attention. As a result of changing theta_r (point 5 of this review) most of the figures were updated.
Fig. 2 (histogram)
Fig. 3 (histogram)
Fig. 6 (C & D)
Fig. 8
Fig. 9
Fig. 10 (added see point 1 from this review)
Fig. C1 (Fig 10 of previous draft now in appendix C)

We have also updated the "code and data availability" section with doi references.

Some remarks about the response-to-review document:

1) Particularly, I think that the test of switching on/off the compaction routine is very insightful, and the comparison of Fig. A and B in the response document should end up in the final manuscript in my opinion. First, Fig. A suggests that the numerical implementation of Richards equation is behaving numerically stable, and second, the comparison of Fig. A and B shows clearly where the next developments should be. This is not only for the CROCUS team, but also for the snow community as a whole: how to accurately describe the effect of liquid water flow on compaction and wet snow metamorphism, as well as how to take care of feedback mechanisms between both. This discussion would just increase the group of researchers for which the manuscript is interesting.

We agree that these two figures (A and B from previous response document)

demonstrate that the Richards routine is behaving as expected and problems arise from the compaction routine. The figure below were added to the revised manuscript, and shows the simulated forcing with SNOWCROMETAMO and SNOWCROCOMPACT turned off. With the correction to theta_r (see point 5 of this document) running Crocus without the compaction and metamorphism routines requires an additional restriction on the time step t to be stable. The maximum time step length was restricted to a maximum of 30 seconds.

[Figure]

Figure shows results from the simulated data set with SNOWCROMETAMO and SNOWCROCOMPACT turned off. Time step t was held below 30 seconds.

Unfortunately running Crocus with SNOWCROMETAMO off and SNOWCROCOMPACT on is no longer stable, after updating theta_r. Hydraulic head magnitudes becomes very large (both positive and negative values occur) which results in errors when computing "CPSI" in the code (the derivative of the water retention curve). The implication of turning off the two routines corresponds to pushing the parameterizations of water retention and hydraulic conductivity beyond the range of physical plausibility. In other words to make these figures, water is passing through dendritic powder snow in Fig A, and through "compacted dendritic powder snow", which is not physically possible in Fig B (of previous review reply).

Sections 5.4 and 6.4 have been added to the revised manuscript to introduce the experiment of turning off feedback and discus the above figure.

2) Related to the previous issue: It's very clear now that the unrealistic alternating pattern arises from the interaction of the water percolation routine and the snow settling. As far as I now understand it, the problem is that higher values of LWC are typically associated with stronger settling rates. This leads to more snow compaction, higher densities, which in turn results into denser packing of the snow crystals, higher capillary suction, and thus, larger values of LWC, which in turn increases the snow

settling, etc. Am I correct here? If the authors agree with this reasoning, I think it is a good idea to explicitly explain it in the manuscript (it is now rather implicit). Actually, an obvious, and easy solution to this problem could be to limit the LWC used in the parameterizations for snow compaction and snow metamorphism to the 5% from the bucket scheme, with the motivation that this is done because the parameterizations for settling and metamorphism are typically developed and tested using the bucket scheme, and future studies may address the behaviour of snow with very high LWC. Did the authors tried this? I strongly recommend to try this out and report the results in the manuscript. If it doesn't solve the problem, it shows that the problems are of more substantial nature.

We believe that our message was not formulated clear enough and the feedback system was only partly understood. The feedback you described is correct but gets further complicated with the inclusion of hydraulic conductivity, the metamorphism and grid re-meshing when a layer melts.

A more compact snow layer will not always result in higher suction, which can be seen in Figure 2 of the revised manuscript. Compare the "decomposed" curve (200 kg/m3, 0.5 mm which closely follows the melt forms curve) with the "small round" curve (400 kg/m3, 0.5 mm). This shows that a layer with higher density has higher suction then less dense snow at low saturations (<12% saturation), but a lower suction (>12% saturation) at high saturations.

Similar behavior is seen in Figure 3 with the hydraulic conductivity curves. More dense snow has a larger hydraulic conductivity for low saturations (<~4% for small rounds & decomposed example), and at higher saturations less dense snow has a larger hydraulic conductivity.

The same behavior can be applied to grain size by using small round and melt forms which have the same density in figures 2 and 3 (revised manuscript).

These two feedback mechanisms alone cannot explain the striped pattern. The stripes suggest that there is a spatial aspect to the feedback, where one snow layer affects the neighboring layers. The method used for solving the Richards equation, where the average value of hydraulic conductivity and suction on the snow layers interfaces with iterations, can start to explain where the striped pattern arises. A single layer interface that has strong positive feedback will affect all the layer interfaces over the course of many iteration. However this issue is not yet fully understood.

We haven't been able to pinpoint the exact cause for the stripes, but through extensive testing outlined the mechanisms responsible for this behavior, i.e. turning off the compaction routine stops the stripes. Restricting the LWC to 5% for the compaction routine was unsuccessful at reducing the stripes, see figure below. Compaction is based on mass of overlying snow and viscosity. We imposed a 5% pore space LWC

limit on the wet deformation rate calculation leaving the total mass of the overlying snow with the water amount that the Richard routine uses (could be >5%). Finding the exact mechanisms producing these undesired stripes and potential remedies will remain subject for future research.

[Figure]

Figure shows results from restricting LWC at the C13 meatmorphsim routine the wet snow viscosity in the compaction routine to the bucket routines limit of 5% volume of pores space. Mass of water in compaction routine was not restricted to 5%.

We have added section 5.4 and 6.4 on the feedbacks that exist between the parameterizations and the compaction and metamorphism routines.

3) I asked to mention the CPU time needed somewhere in the manuscript. I do understand the point of the authors that many factors influence the CPU time, as for example the file output resolution. However, I still think it is very important for such model/numerics description papers to give readers an idea of the computational burden of the proposed model/numerics improvements. So maybe the authors can just provide the relative extra CPU time to use the new Richards equation scheme over the old bucket scheme, or at least state that the CPU time needed for the Richards equation scheme is of the same order of magnitude as the bucket scheme, as it seems to be. Note that it is important to consider that a switch to 1-dimensional Richards equation is generally accompanied by a very acceptable increase in computational time, in contrast to the three-dimensional snow models recently proposed (for example by Hirashima et al. 2014, Leroux and Pomeroy (2017)), which are accompanied by such a large increase of computational time, that a useful application on seasonal time scales or large spatial scales on real natural snowpacks is not (yet) feasible.

We added a paragraph at the start of the discussion (section 6) that compares CPU time of Richards and the bucket routines.

4) I made the remark that a plot of snowpack runoff is very useful. The in response

provided plot of the soil moisture in the upper soil layer is also interesting, but not so informative when it comes to snowpack runoff. The reason I asked about runoff is that in snow modelling, many researchers are interested in the hydrological aspects, mostly snowpack runoff. It should just be easy to plot it, as it results directly from the free-drainage boundary condition (just translate gradient in pressure head to the flux). Two important aspects that I hope show up is a shift later in the day of the arrival of the meltwater at the bottom of the snowpack, and a recession curve at night. This aspect is not a show-stopper for me, but it would just enhance the impact of the manuscript.

Here is a plot on the flux from the bottom of the snowpack for day 89 of the simulated data set, which is the next to last day of the simulation. There is no flux to the soil on day 89 of the simulated data set for the bucket routine. The bucket routine only passes water on the last few time steps of the simulation, and therefore we do not have a good reference for soil flux with the bucket routine.

[Figure]

We have not included this to the manuscript because we do not have validation data to compare this with. We cannot assess if there is a shift in peak water flux from the plot above but Fig. 8 of the revised manuscript shows a delay in the diurnal water fronts through the snowpack. We do notice a decay of the water flux during the night.

5) Fig. A in the response-to-review document is very important and I strongly recommend to take it into the manuscript. However, Fig. A shows one confusing thing: after passing of the melt water front, a significant amount of liquid water (typically 2-3%) should be held in the capillaries, although the plot seems to suggest that the water content after passing of the melt water front falls back to almost 0 in some layers. One suspicion I have is that Eq. 7 is not implemented correctly. As far as I understand the code, Eq. 7 from the manuscript translates into the following source code line:
ZTHETA_R= MIN(ZTHETA*.75, 0.02)

However, this is not consistent with Eq. 7. It should translate into:
IF(ZTHETA.LT.0.02)THEN
ZTHETA_R=0.75*ZTHETA
ELSE
ZHETA_R=0.02
ENDIF

The latter approach is in my opinion the better one. Otherwise, a condition can occur that even when theta>0.02, theta_r is reduced below 0.02, after which theta gets smaller, after which theta_r is reduced, and this continues all the way to 0, although we know that a bulk liquid water content for a wet snowpack is typically more than 2%. Note that for numerical stability, it may be better to write something like IF(ZTHETA.LT.(0.02 + theta_min)), such that ZTHETA is always significantly larger than ZTHETA_R, which is required in the van Genuchten model.

This has been corrected and the figures in the manuscript have been updated to reflect this. This correction keeps snow layers above theta_r after the first substantial wetting. The conclusions drawn (about feedback) using "min theta_r" are not affected by this correction. We found that theta_min could be changed form $10^{-6}$ to $10^{-5}$ without changing the timing of the warming front (figure C1 and appendix C of the revised manuscript), so the default value has been changed, because $10^{-5}$ makes for a slightly faster CPU run time.

6) The discussion about the time steps inside the Richards equation solver in combination with the mass balance check, as provided in the response-to-review document, should be present in the manuscript I think, in a much more condensed form of course. It is important to note in the manuscript that the mass balance error is acceptable. Otherwise, I can imagine that other readers will also feel that the allowed mass balance error is large compared to the minimum allowed time step.

We added some sentences to Appendix A where the mass balance error is discussed.
Some remarks about the manuscript:

7) Section 6.2.2: I think the authors are too negative about the free-flow bottom boundary (p13,l10-11). First, in most snow models, the outflow from the snowpack is not at all constrained by the underlying soil. The bucket approach in CROCUS is probably also not taking into account the conditions of the underlying soil. One can always argue that the soil module of ISBA should take care of the incoming water flux from the snowpack, i.e., decide if it infiltrates into the soil, or creates overland flow. Note that in reality, a frozen, saturated, or extremely dry soil can have such a reduced infiltration capacity, that meltwater from the snowpack creates a significant amount of lateral overland flow, and thereby constitutes a significant flood risk. I agree with the authors that the approach of SNOWPACK to solve the snow-soil continuum at once has the advantage that these processes can be adequately captured, and with the SNOWPACK model, we are indeed able to reproduce melt pond formation from snowpack runoff in case the soil has limited infiltration capacity. On the other hand, one looses the sophisticated coupling some hydrological models have from the unsaturated zone to the aquifer and streamflow (I'm not sure how this is with ISBA). I think in section 6.2.2., the authors may want to discuss some of these aspects there.

Section 6.2.2 has been updated and is less negative, however one of the major motivations for this study by the Crocus team was to better couple the soil and snowpack routines.

8) Prewetting amount: If I understand correctly, Fig 8 A and D should be identical with Fig. 10 A and B, as both have the same prewetting amount of 10^-5? Yet, there is a clear difference, but it is not clear where this originates from. Is it a typo in the caption of Fig 8, that it should actually be 10^-6? Or what else changed between both simulations? Another issue here: sections 5.3 and 6.4 now fail to explain why the prewetting is so important, but I think the reason is that when it is set too high, hydraulic conductivity becomes already so significant, that water percolates, even when we should still consider the snowpack "dry". When more water is added after the prewetting and this is refrozen every time step, heat is advected. If authors agree with this explanation, they may consider adding it to the manuscript. Note that in SNOWPACK, we don't refreeze prewetting water every time step. We have hysteresis, i.e., the threshold for executing phase changes is a factor 10 larger than the prewetting amount. I think a similar approach in CROCUS would help to reduce the warming effect. But I consider this something for future work. In my opinion, the discussion of the sensitivity of the pre-wetting amount is not so interesting, and as I suggest to add figures describing the influence of switching on/off the compaction routines, this section and Fig. 10 could be removed to save space, if the authors wish to do so.

Yes there is a typo here. We did not want to include identical figures so the figures should be one order of magnitude above and below that what we pick as default. We think that the pre-wetting amount is still important as it is a major difference between

solving Richards equation in soil and snow. Pre-wetting amount is also an unphysical parameter that was used so we feel that it is important to show a sensitivity test on it. Since the results of the sensitivity test do not show that the variable is sensitive below a threshold value, we chose to include this in Appendix C.

Technical corrections (line numbers refer to revised manuscript):
(Note that I think that some of the technical corrections should have been identified before submission by a proper proofreading by author and co-authors.)
- p1,l11: "thought" -> "through"
This has been changed in the revised manuscript.
- p1,l12: add gravity: "capillary suction, gravity and hydraulic conductivity"
This has been changed in the revised manuscript.
- p1,l15: "coved" -> "covered" (?)
This has been changed in the revised manuscript.
- p1,l22: "The parameterization ... crust layers." This is an assumption (although a well justified one) and not really supported by data in the manuscript, so I think this sentence is misplaced in the abstract.
This has been taken out of the abstract.
- p1,l30: either remove comma or write: "time consuming, and LWC"
This has been changed in the revised manuscript.
- p1, l30-31: "change over timescales that are"
This has been changed in the revised manuscript.
- p2, l2: remove comma: "rescue workers have reported"
This has been changed in the revised manuscript.
- p2, l26: "due to suction and sloping terrain and water pooling"
This has been changed in the revised manuscript.
- p2, l29: "and heterogeneous" -> "as well as heterogeneous"
This has been changed in the revised manuscript.
- p3, l22: "however" -> "although"
This has been changed in the revised manuscript.
- p5, l2: "found the speed"
This has been changed in the revised manuscript.
- p6, l12: note that Eq. 7 is not a continuous function, there is a break at theta=0.02. Either make the function continuous, or say "a piecewise function".
This has been changed in the revised manuscript.
- p6, l17: I suggest to write: "which corresponds to a minimum pressure head, for which it holds for every dry snow layer that the liquid water content is smaller than a prescribed minimum value \theta_min."
This has been changed in the revised manuscript.
- p6: Eq. 8 and 9 uses different symbols to indicate multiplication. The "x" is not adequate for scalar multiplication.
This has been changed in the revised manuscript.
- p6, l28: add comma "is a complex system, it is"

This has been changed in the revised manuscript.
- p7, l2: Eq 8 should refer to Eq. 10? (Occurs twice)
This has been changed in the revised manuscript.
- p7, l7: add comma "computations, the following"
This has been changed in the revised manuscript.
- p7, l8: I suggest "There must be a substantial snowpack: if there are less than 3 layers"
This has been changed in the revised manuscript.
- p7, l9: The sentence is incomplete.
This has been changed in the revised manuscript.
- p7, l26: "borders"
This has been changed in the revised manuscript.
- p8, l25: I would move the sentence "The lower boundary is the soil-snow interface." Before starting the paragraph: "There are two options". Otherwise it is not clear that the bottom boundary condition refers to the snowpack, and not the soil.
This has been changed in the revised manuscript.
- p9, l2: Eq. 10 should point to Eq. 11?
This has been changed in the revised manuscript.
- p9, l24: For clarity, I suggest: "The peak shortwave radiation"
This has been changed in the revised manuscript.
- p10, l20-21: This is not so clear. I suggest: "causes the surface layers to get wet, while deeper layers remain below freezing".
This has been changed in the revised manuscript.
- p10, l24: "shows the formation"
This has been changed in the revised manuscript.
- p11, l4: I would write: "pore space that is filled by water (i.e., the saturation) ...", as "saturation" is the term used in the figure.
This has been changed in the revised manuscript.
- p11, l21-22: This sentence is not grammatically correct. Should be something like ".. is drastically different .., although the timing ..."
This has been changed in the revised manuscript.
- p12, l4-5: "Routines such as for compaction and grain metamorphism were ..."
This has been changed in the revised manuscript.
- p12, l14-20: I guess the authors do not plan to print this in bold.
This has been changed in the revised manuscript.
- p12, l18: "modles"
This has been changed in the revised manuscript.
- p12, l25: I would start this sentence with: "When using the upper soil layer as lower boundary, the hydraulic conductivity" to make clear that this is in contrast to the free-flow boundary.
This has been changed in the revised manuscript.
- p13, l8: "there is there is"
This has been changed in the revised manuscript.
- p15, l6: add comma "snow layer, snow"

This has been changed in the revised manuscript.
- p15, l8: I think a nicer formulation would be: "are the only crystal type that needs to be described by a water retention curve."
This has been changed in the revised manuscript.
- p15, l13: note that a recent study shows the effect of hysteresis in snow, a study that I recommend citing here: Leroux and Pomeroy (2017).
This has been added to the revised manuscript.
- p16, l5: "The parameterizations"
This has been changed in the revised manuscript.
- p15, l7 and p16, l9. Note that the abbreviation MF is not introduced, but as it is only used twice, I would recommend to write "melt forms" in both cases and not use an abbreviation.
This has been changed in the revised manuscript.
Fig 9 and 10: Make the captions more in line with Fig. 8, for example: "Crocus output for Neverland forcing using different time steps ..."
This has been changed in the revised manuscript.
Fig 10: Subfigure B has different label, and wrongly formatted. I prefer the B style, that the labels show everyhwere "theta_min = XXXX"
This has been changed in the revised manuscript.
Fig 10: The caption only indicates what A and B refer to, but not C and D, and according to the figure itself B is also with pre-wetting 10^-5 and not 10^-7 as is written in the caption.
This has been changed in the revised manuscript.

References:
Hirashima, H., Yamaguchi, S., and Katsushima, T.: A multidimensional water transport model to reproduce preferential flow in the snowpack, Cold Reg. Sci. Technol., 108, 80-90, doi:10.1016/j.coldregions.2014.09.004, 2014.
Nicolas R. Leroux, John W. Pomeroy, Modelling capillary hysteresis effects on preferential flow through melting and cold layered snowpacks, Advances in Water Resources, Volume 107, 2017, Pages 250-264, ISSN 0309-1708, http://dx.doi.org/10.1016/j.advwatres.2017.06.024.